# Investigating a Water Resource Allocation Model by Using Interval Fuzzy Two-Stage Robust Planning for the Yinma River Basin, Jilin Province, China

**Hao Zhang** [1,†], **Wei He** [1,†], **Haihong Xu** [2,*], **Hao Yang** [1], **Zhixing Ren** [3], **Luze Yang** [4], **Peixuan Sun** [4], **Zhengyang Deng** [5], **Minghao Li** [1], **Shengping Wang** [1,6,*] and **Yu Li** [1,*]

1   MOE Key Laboratory of Resources Environmental Systems Optimization, North China Electric Power University, Beijing 102206, China; zhanghao@ncepu.edu.cn (H.Z.); 120202232011@ncepu.edu.cn (W.H.); 120212232080@ncepu.edu.cn (H.Y.); lmh66@ncepu.edu.cn (M.L.)
2   Appraisal Center for Environmental & Engineering Ministry of Ecology and Environment, No. 28 Beiyuan Road, Beijing 100012, China
3   College of Forestry, Northeast Forestry University, Harbin 150040, China; RenzhixingRyy@outlook.com
4   College of New Energy and Environment, Jilin University, Changchun 130012, China; yanglz19@mails.jlu.edu.cn (L.Y.); sunpx19@mails.jlu.edu.cn (P.S.)
5   College of Resources and Environment, Jilin Agricultural University, Changchun 130118, China; 20190299@mails.jlau.edu.cn
6   College of Hydraulic and Hydro-Power Engineering, North China Electric Power University, Beijing 102206, China
*   Correspondence: xuhh@acee.org.cn (H.X.); wangshp418@126.com (S.W.); liyuxx@ncepu.edu.cn (Y.L.); Tel./Fax: +86-10-6177-2836 (Y.L.)
†   These authors have contributed equally to the study and they receive equal credit.

**Abstract:** This study introduces a fuzzy method to construct the interval fuzzy two-stage robust (ITSFR) water resource optimal allocation model based on the interval two-stage robust (ITSR) water resource optimal allocation model. Optimal economic benefit was considered the objective function, and the number of available water resources, sewage treatment capacity, reuse water treatment capacity, and total pollutant control were considered as the constraints. Under three five-year planning periods (2015–2020, 2020–2025, and 2025–2030) and according to the allocation levels of dry, flat, and abundant water periods (low, medium, and high discharge), the pollution absorption, upgrading projects, and water resource allocation schemes of various water sectors (industry, municipal life, ecological environment, and agricultural sector) in the Yinma River Basin were optimized. Water consumption quota is an interval value; high and low water consumption lead to a waste of water resources in the water consumption sector and restrict the development of the water consumption sector, respectively, which indicates that the water consumption quota has the characteristics of fuzzy uncertainty. Therefore, the optimization model was set as a fuzzy parameter in the solution process. The simulation results indicated that water quota can directly influence the income of water resource use, and thus, indirectly influence the economic benefit of the Yinma River Basin during the planning period. In the planning period of the Yinma River Basin, the economic benefit interval of dry, flat, and abundant water periods was reduced by 57%, 55%, and 48%, respectively, which provides a robust method with the advantages of a balanced economy, a stable system, reduced decision-making space, and significantly improved decision-making efficiency. Moreover, the emission ranges of typical pollution indicators (chemical oxygen demand (COD) and ammonia nitrogen) in the eight counties and urban areas of the Yinma River Basin were significantly reduced during the three planning periods (Dehui area had the highest overall reduction of ammonia nitrogen in the industrial sector during the second five-year planning period, up to 65%), which indicated a significant improvement in the decision-making efficiency. In addition to the Changchun City planning areas dominated by the agriculture production water sector, water resource allocation accounts for >80% of the regional water resource allocation; using the fuzzy optimization method after the Yinma River Basin water resource allocation model, the overall water deficit was significantly reduced; moreover, it was almost the same as in the first five-year period of Changchun City industry

water deficit, which declined by up to 33%. The problem of resource waste caused by excessive water limiting in the water sector could be avoided because of the fuzzy water limit. To solve the prominent problem of water deficit in large- and medium-sized cities in the basin, industrial and ecological water sectors can implement measures such as water resource reuse. The total amount of water reuse in a medium year increases by up to 46% compared with that in the ITSR optimization model, which can be attributed to the reduced water consumption limit range of water consumption sectors after the fuzzy water consumption limit. This shows that more water can be allocated to meet the requirements of the water sector during decision-making. In conclusion, this study offers an effective scheme for decision makers to plan water resource allocation in the Yinma River Basin.

**Keywords:** interval fuzzy two-stage robust optimization method; sewage lifting project; water allocation; Yinma River Basin

## 1. Introduction

Rapid economic growth and population density have markedly increased the water consumption and wastewater discharge levels. The long-term and large-scale discharge of pollutants damages the self-purification ability of water bodies, and poor river quality in turn may affect human health and the environment, leading to the loss of biodiversity. Freshwater system pollution is a global environmental problem; therefore, the protection of water bodies is essential to prevent the further deterioration of environmental quality. Measures to prevent water pollution will benefit the ecosystem and human health [1–3]. Industrial wastewater is the largest discharge source of toxic pollutants and the third largest discharge source of nutrients in China's water bodies. Due to insufficient rural sewage treatment facilities and poor sewage treatment capacity, waste deposition on rural residential land negatively impacts the river water quality [4–6]. Zou et al.'s study demonstrated that the reduction of COD and NH3-N levels shares a power function relationship with the pollutant emission reduction ratio energy consumption; the higher the pollutant emission reduction, the lower the pollutant emission ratio energy consumption [7]. Najafzadeh et al. [8] used multivariate adaptive regression spline (MARS) and least square support vector machine (LS-SVM) as machine learning methods to predict five-day biochemical oxygen demand (BOD5) and chemical oxygen demand (COD). Najafzadeh et al. [9] used gene expression programming (GEP), evolutionary polynomial regression (EPR), and model tree (MT) methods to estimate the three indicators of biochemical oxygen demand (BOD), dissolved oxygen (DO), and chemical oxygen demand (COD). Due to the steadily increasing demand for clean water, increasingly better powerful strategies should be implemented to reduce water pollution, and the use of key chemicals should be reduced to decrease their discharge into the environment; moreover, for a reasonable allocation of water resources, highly effective and economical means of improving the pollutant-carrying capacity should be developed to address the existing pollution problem [10–12].

The Yinma River Basin, located in Jilin Province, China, serves as a crucial water source for domestic and farmland irrigation purposes in this region, and it is a major tributary of the Songhua River system. With rapid urbanization and industrial and economic developments in recent years, pollution in the Yinma River Basin due to industries, pharmaceuticals, aquaculture, and animal husbandry has increased. The main sources of pollution are agricultural, industrial, and residential activities, which involve the use of chemical fertilizers and pesticides and the discharge of industrial wastewater and domestic sewage. The improper protection and utilization of water resources have increased water environment risks in this region [13–15]. Reducing water resource pollution and optimizing water resource allocation can positively impact the development of watershed. Optimizing water resource allocation is essential for establishing a robust and efficient water resource management system. The formulation and implementation of water resource allocation planning in the Basin is an effective measure to promote the orderly development, efficient

utilization, management, and rational allocation of water resources [16,17]. A reasonable allocation of water resources can achieve economic optimization by ensuring the sustainable development of resources [18,19].

Two-stage stochastic programming can modify a predetermined target according to the overall impact of uncertain events (first-stage decision-making) and generate corresponding decisions after the occurrence of an uncertain event (second-stage decision-making) [20]. Interval linear programming characterizes the parameters and results in an uncertain system in the form of interval numbers and replaces the deterministic values with interval numbers [21]. A robust optimization method balances the relationship between a variable random quantity and a system risk to achieve economic benefit optimization [22]. Fuzzy programming can effectively deal with optimization problems with non-randomness, inaccuracy, and fuzziness [23]. Li et al. [24] applied a fuzzy-two-stage-robust method to study regional air pollution control and management. Zhen et al. [25] applied an interval-two-stage-robust method to optimize Tangshan regional power system management. The interval-fuzzy-two-stage robust (ITSFR) method has also been applied for the long-term management planning of municipal solid waste, water resource management under uncertain conditions, and agricultural production scale planning under limited water resource conditions [20,26,27]. Meng et al. and He et al. applied interval-two-stage and interval-two-stage-robust methods, respectively, to optimize water resource allocation in the Yinma River Basin. However, the fuzzy uncertainty of relevant parameters has not been effectively considered, and the decision space is not conducive to the decision planning of planners [17,22].

Based on existing research, this study constructed a water resource allocation model of the Yinma River Basin based on the interval-fuzzy-two-stage-robust (ITSFR) optimization method. In the model, the optimal economic benefit of the basin was considered the objective function and water resource revenue and cost were considered the constraints. Considering the fuzzy uncertainty of parameters such as departmental water quota to avoid waste caused by inefficient allocation of water resources, the proposed model provides insights into the measures to mitigate watershed pollution, enhance economic benefits, formulate reasonable water resource allocation plans, and provide feasible ideas for planning.

## 2. Case Study

Yinma River is the largest tributary of the Second Songhua River, and it is located in the southeast of Laoye Ling, Diyuanzi Township, Yitong County, which is situated in the central part of Jilin Province [28]. This river originates from Hulanling, Yima Town, flows through Panshi City, Shuangyang, Yongji, Jiutai, Dehui, and Nong'an, that is, six districts and counties, to the confluence of Kaoshan Town, Nong'an County, and the Yitong River, flowing nearly 20 km north into the Songhua River, with a total length of 386.8 km and a watershed area of approximately 16,000 km$^2$ (including the Yitong River) [29]. The distribution map of the Yinma River Basin is shown in Figure 1. The river is located in the core area of the black soil belt in northeast China. It is the main production area of grains such as corn and rice, which makes it the main commercial grain production region in China. The river flows through the Shitoukoumen and Xinlicheng Reservoirs, providing water for domestic purposes in the urban region of Changchun and for domestic purposes, agricultural irrigation, and fish farming in the surrounding areas. The basin has a typical continental monsoon climate in the north temperate zone; the average annual rainfall in this region is far less than the rate of evaporation, and it is mainly concentrated in the summer [30,31]. The uneven distribution of water resources in the Yinma River Basin affects the residents in the region and restricts local economic development.

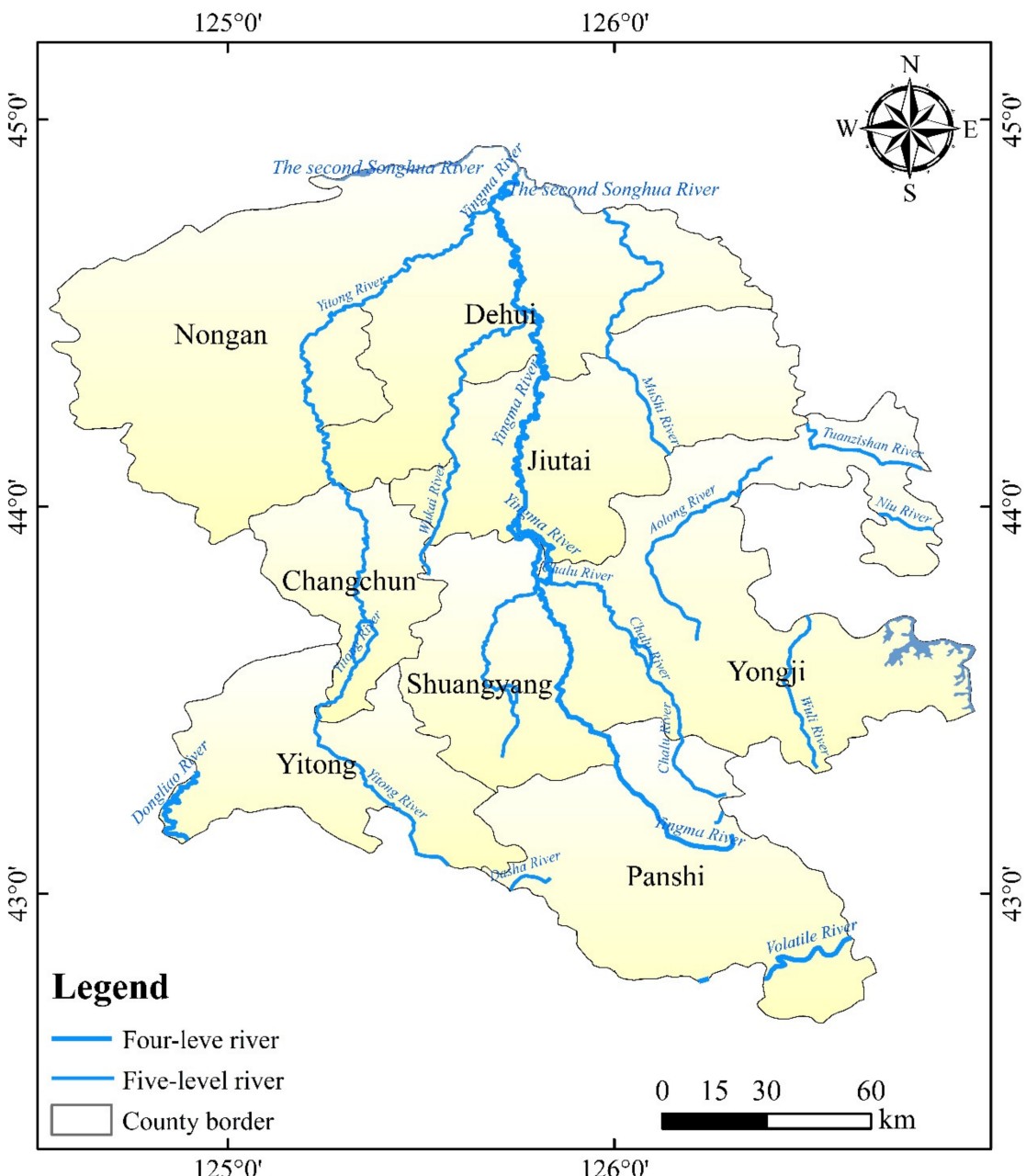

**Figure 1.** Distribution map of the Yinma River Basin.

## 3. Model Formulation

### 3.1. Constructing a Water Resource Allocation Model in the Yinma River Basin Based on the ITSFR Optimization Method

In the water resource allocation model based on the ITSR optimization method of the Yinmahe River Basin [22], different water consumption departments determine the highest and lowest water quota in the model by considering calculated values of its upper and lower limits; however, these departments did not consider the water quota for interval values and uncertainty in decision-making. Therefore, in this study, the fuzzy method was incorporated based on the ITSR model to construct a water resource allocation model of the Yinma River Basin based on the ITSFR optimization method. The maximum and minimum water consumption quota limits of water consumption departments in different regions were unclear. Depending on the pollution capacity improvement project implemented, to optimize water resource allocation in the Yinma River Basin, the developmental needs of different water consumption departments in different regions and the constraint conditions

of the Basin's sewage-carrying capacity were fully considered. The model description is as follows:

Objective function:

$$\max \lambda^{\pm} \tag{1}$$

Constraints:

1. Economic benefit optimization constraint:

$$f_1^{\pm} - f_2^{\pm} - f_3^{\pm} - f_4^{\pm} - f_5^{\pm} - f_6^{\pm} - f_7^{\pm} \geq f_z^+ - (1 - \lambda^{\pm}) \cdot (f_z^+ - f_z^-) \tag{2}$$

Income from water use:

$$f_1^{\pm} = \sum_{j=1}^{8} \sum_{k=1}^{4} \sum_{t=1}^{3} L_t \cdot UNB_{jkt}^{\pm} \cdot \left( IAW_{jkt}^{\pm} + RW_{jkt}^{\pm} \right) - \sum_{j=1}^{8} \sum_{k=1}^{4} \sum_{t=1}^{3} \sum_{h=1}^{3} L_t \cdot p_h \cdot PNB_{jkt}^{\pm} \cdot DW_{jkth}^{\pm} \tag{3}$$

Cost of water use:

$$f_2^{\pm} = \sum_{j=1}^{8} \sum_{k=1}^{4} \sum_{t=1}^{3} L_t \cdot \left( IAW_{jkt}^{\pm} - \sum_{h=1}^{3} p_h \cdot DW_{jkth}^{\pm} \right) \cdot CW_{jkt}^{\pm} \\ + \sum_{j=1}^{8} \sum_{k=1}^{4} \sum_{t=1}^{3} L_t \cdot RW_{jkt}^{\pm} \cdot CRW_{jkt}^{\pm} \tag{4}$$

Sewage treatment cost:

$$f_3^{\pm} = \sum_{j=1}^{8} \sum_{k=1}^{4} \sum_{t=1}^{3} L_t \cdot \left( \begin{array}{c} IAW_{jkt}^{\pm} - \sum\limits_{h=1}^{3} p_h \cdot DW_{jkth}^{\pm} \\ + RW_{jkt}^{\pm} \end{array} \right) \cdot \alpha_{jkt}^{\pm} \cdot CWW_{jkt}^{\pm} \\ + \sum_{j=1}^{8} \sum_{k=1}^{4} \sum_{t=1}^{3} L_t \cdot RW_{jkt}^{\pm} \cdot CRWT_{jkt}^{\pm} \tag{5}$$

Pollutant-carrying capacity increases project cost:

$$f_4^{\pm} = \sum_{i=1}^{11} \sum_{l=1}^{7} \sum_{t=1}^{3} ER_{ilt}^{\pm} \cdot Y_{ilt}^{\pm} \cdot CER_{ilt}^{\pm} \tag{6}$$

Control the cost of punishment:

$$f_5^{\pm} = \rho \sum_{j=1}^{8} \sum_{k=1}^{4} \sum_{t=1}^{3} \sum_{h=1}^{3} L_t \cdot p_h (PNB_{jkt}^{\pm} \cdot DW_{jkth}^{\pm} - p_h \sum_{h=1}^{3} PNB_{jkt}^{\pm} \cdot DW_{jkth}^{\pm} + 2\theta_h^{\pm}) \tag{7}$$

$$f_6^{\pm} = \rho \sum_{j=1}^{8} \sum_{k=1}^{4} \sum_{t=1}^{3} \sum_{h=1}^{3} L_t \cdot p_h (CW_{jkt}^{\pm} \cdot DW_{jkth}^{\pm} - p_h \sum_{h=1}^{3} CW_{jkt}^{\pm} \cdot DW_{jkth}^{\pm} + 2\theta_h^{\pm}) \tag{8}$$

$$f_7^{\pm} = \rho \sum_{j=1}^{8} \sum_{k=1}^{4} \sum_{t=1}^{3} \sum_{h=1}^{3} L_t \cdot p_h \left( \begin{array}{c} \alpha_{jkt}^{\pm} \cdot CWW_{jkt}^{\pm} \cdot DW_{jkth}^{\pm} \\ - p_h \sum\limits_{h=1}^{3} \alpha_{jkt}^{\pm} \cdot CWW_{jkt}^{\pm} \cdot DW_{jkth}^{\pm} + 2\theta_h^{\pm} \end{array} \right) \tag{9}$$

where $j$ = 1 to 8 represents the 8 counties and cities that the Yinma River flows through; $k$ = 1, 2, 3, and 4 represents industry, municipal life, ecological environment, and agriculture, respectively; $t$ = 1, 2, and 3 are the first (2015–2020), second (2020–2025), and third planning periods (2025–2030), respectively; $r$ = 1 and 2 represent COD and ammonia nitrogen, respectively; $l$ = 1–7 represent wetlands, floating beds, corridors, pre-storehouses, conservation forests, silt removal, and aeration, respectively; $\rho$ is the robust coefficient, the values are 0, 0.8, and 1.

2. Constraints on available water resources [22]:

$$\sum_{k=1}^{4}\left(IAW_{jkt}^{\pm} - DW_{jkth}^{\pm}\right) \le AWQ_{th}^{\pm}; \ \forall t, h \tag{10}$$

$$DW_{jkth}^{\pm} \le IAW_{jkt}^{\pm}; \ \forall j, k, t, h \tag{11}$$

3. Constraints on departmental water resources demand:

$$IAW_{jkt}^{\pm} - DW_{jkth}^{\pm} + RW_{jkt}^{\pm} \ge WD_{\min jkt}^{+} - (1 - \lambda^{\pm}) \cdot (WD_{\min jkt}^{+} - WD_{\min jkt}^{-}); \ \forall j, k, t, h \tag{12}$$

$$IAW_{jkt}^{\pm} - DW_{jkth}^{\pm} + RW_{jkt}^{\pm} \le WD_{\max jkt}^{-} + (1 - \lambda^{\pm}) \cdot (WD_{\max jkt}^{+} - WD_{\max jkt}^{-}); \ \forall j, k, t, h \tag{13}$$

4. Constraints on sewage treatment capacity [22]:

$$\sum_{k=1}^{2}\left(IAW_{jkt}^{\pm} - DW_{jkth}^{\pm} + RW_{jkt}^{\pm}\right) \cdot \alpha_{jkt}^{\pm} \le ATW_{jkt}^{\pm}, \forall j, k, t, h \tag{14}$$

5. Constraints on recycling water treatment capacity [22]:

$$\sum_{k=1}^{2}\left(IAW_{jkt}^{\pm} - DW_{jkth}^{\pm} + RW_{jkt}^{\pm}\right) \cdot \alpha_{jkt}^{\pm} \cdot \xi_{jkt} \ge \sum_{k=1}^{4} RW_{jkt}^{\pm}, \forall j, t \tag{15}$$

6. Total pollutant control constraints [22]:

$$\sum_{k=1}^{4}\left(IAW_{jkt}^{\pm} - DW_{jkth}^{\pm} + RW_{jkt}^{\pm}\right) \cdot \alpha_{jkt}^{\pm} \cdot \beta_{jkt}^{\pm} \cdot EC_{krt}^{\pm} \le TED_{jrt}^{\pm}, \forall j, r, t, h \tag{16}$$

7. Constraints on basin pollutant-carrying capacity [22]:

$$\sum_{j=1}^{8}\sum_{k=1}^{4}\left(\begin{matrix} IAW_{jkt}^{\pm} - DW_{jkth}^{\pm} \\ + RW_{jkt}^{\pm} \end{matrix}\right) \cdot \alpha_{jkt}^{\pm} \cdot \beta_{jkt}^{\pm} \cdot EC_{krt}^{\pm} \cdot IDR_{krt} \cdot X_{ij} \\ - \sum_{l=1}^{7} EER_{ilrt}^{\pm} \cdot ER_{ilt}^{\pm} \cdot Y_{ilt}^{\pm} \le ALD_{irth}^{\pm}, \forall i, r, t, h \tag{17}$$

8. Non-negative constraint [22]:

$$DW_{jkth}^{\pm}, RW_{jkt}^{\pm}, ER_{ilt}^{\pm} \ge 0 \tag{18}$$

9. Robust constraints [22]:

$$PNB_{jkt}^{\pm} \cdot DW_{jkth}^{\pm} - p_h \sum_{h=1}^{3} PNB_{jkt}^{\pm} \cdot DW_{jkth}^{\pm} + 2\theta_h^{\pm} \ge 0; \ \forall j, k, t, h \tag{19}$$

$$CW_{jkt}^{\pm} \cdot DW_{jkth}^{\pm} - p_h \sum_{h=1}^{3} CW_{jkt}^{\pm} \cdot DW_{jkth}^{\pm} + 2\theta_h^{\pm} \ge 0; \ \forall j, k, t, h \tag{20}$$

$$\alpha_{jkt}^{\pm} \cdot CWW_{jkt}^{\pm} \cdot DW_{jkth}^{\pm} - p_h \sum_{h=1}^{3} \alpha_{jkt}^{\pm} \cdot CWW_{jkt}^{\pm} \cdot DW_{jkth}^{\pm} + 2\theta_h^{\pm} \ge 0; \ \forall j, k, t, h \tag{21}$$

*3.2. Model Solving*

The water resource allocation model of the Yinma River Basin based on the ITSR optimization method constructed by He et al. [22] was considered as the original model in the present study; LINGO 18.0 software (LINDO, Chicago, IL, USA) was used to construct the water resource allocation model of the Yinma River Basin based on the ITSFR optimization method. The model has a two-step solution. First, the upper bound model is solved, and based on this solution, the second stage is solved. When solving the lower bound model, an existing value is substituted into the calculation, and the optimal value of 0.8, obtained using the ITSR model, is used for calculating the robustness coefficient; the results obtained by the model are presented in the form of an interval.

## 4. Results and Discussion

*4.1. Change Analysis of System Economic Benefit in the Yinma River Basin Based on the ITSFR Optimization Method*

In this study, the fuzzy programming method was introduced on the basis of the ITSR model to construct the Yinma River Basin water resource allocation model based on the interval fuzzy two-stage-robust optimization method to ensure economic benefit optimization. The robustness coefficient of 0.8 is optimal in the ITSR model. The economic benefits of the Yinma River Basin system after the optimization of the fuzzy planning method are listed in Table 1.

**Table 1.** Economic benefits of the Yinma River Basin under ITSR and ITSFR models ($10^4$ m$^3$/yuan).

| Scenario | ITSR | ITSFR |
|----------|------|-------|
| h = 1 | [84,696.3778, 112,359.3469] | [100,397.5555, 112,272.6977] |
| h = 2 | [93,401.4222, 113,072.7597] | [104,170.1354, 112,985.4470] |
| h = 3 | [99,662.9147, 113,332.1733] | [106,187.6088, 113,244.8058] |

Table 1 shows that the upper and lower limits of economic benefits of the Yinma River Basin are reduced after optimization using the fuzzy planning method. The upper and the lower limits of economic benefits of the ITSFR model are reduced by [15,701.18, 86.65] $\times$ $10^4$ m$^3$ at a low-flow level compared with those of the ITSR model. At a medium-flow level, the upper and lower limits of the ITSFR model's economic benefits are reduced by [10,768.10, 87.31] $\times$ $10^4$ m$^3$ compared with those of the ITSR model. At a high-flow level, the upper and lower limits of the ITSFR model's economic benefits are reduced by [6524.69, 87.37] $\times$ $10^4$ m$^3$ yuan compared with those of the ITSR model. This is because the ITSR model mainly considers the balance between economy and stability [32,33], although it fails to fully consider the impact of decision scope on decision makers in actual decision-making. The fuzzy programming method can not only ensure balance between economy and stability but also effectively reduce the decision-making space, which ensures that decision makers can plan more efficiently according to the actual situation during decision-making.

*4.2. Total Pollution Target Control in the Yinma River Basin Based on the ITSFR Optimization Method*

The capacity of the water environment refers to the ability of a water body to control the concentration of pollution indicators within a standard range through its own dilution and purification, which reflects the carrying capacity of pollution indicators in the water body [10]. Excess pollution occurs in an aquatic environment when the total amount of pollution indicators discharged into the river exceeds the capacity of that water body; total capacity control, with the capacity of the water environment as the limit, is an important means of preventing and controlling water environment pollution. As important indicators for characterizing the water environment in the Yinma River Basin, pollution indicator discharge and pollution indicator inflow can be considered total pollution control indicators, both of which influence the development and utilization of water resources in the Yinma River Basin.

Resetting the allocation of water resources to ensure optimization can also ensure better control of the total amount of pollution indicators in water bodies and increase the pollutant-carrying capacity of water bodies in the Yinma River Basin. The COD and ammonia emissions of various water-use sectors in the Yinma River Basin during different planning periods are shown in Figure 2.

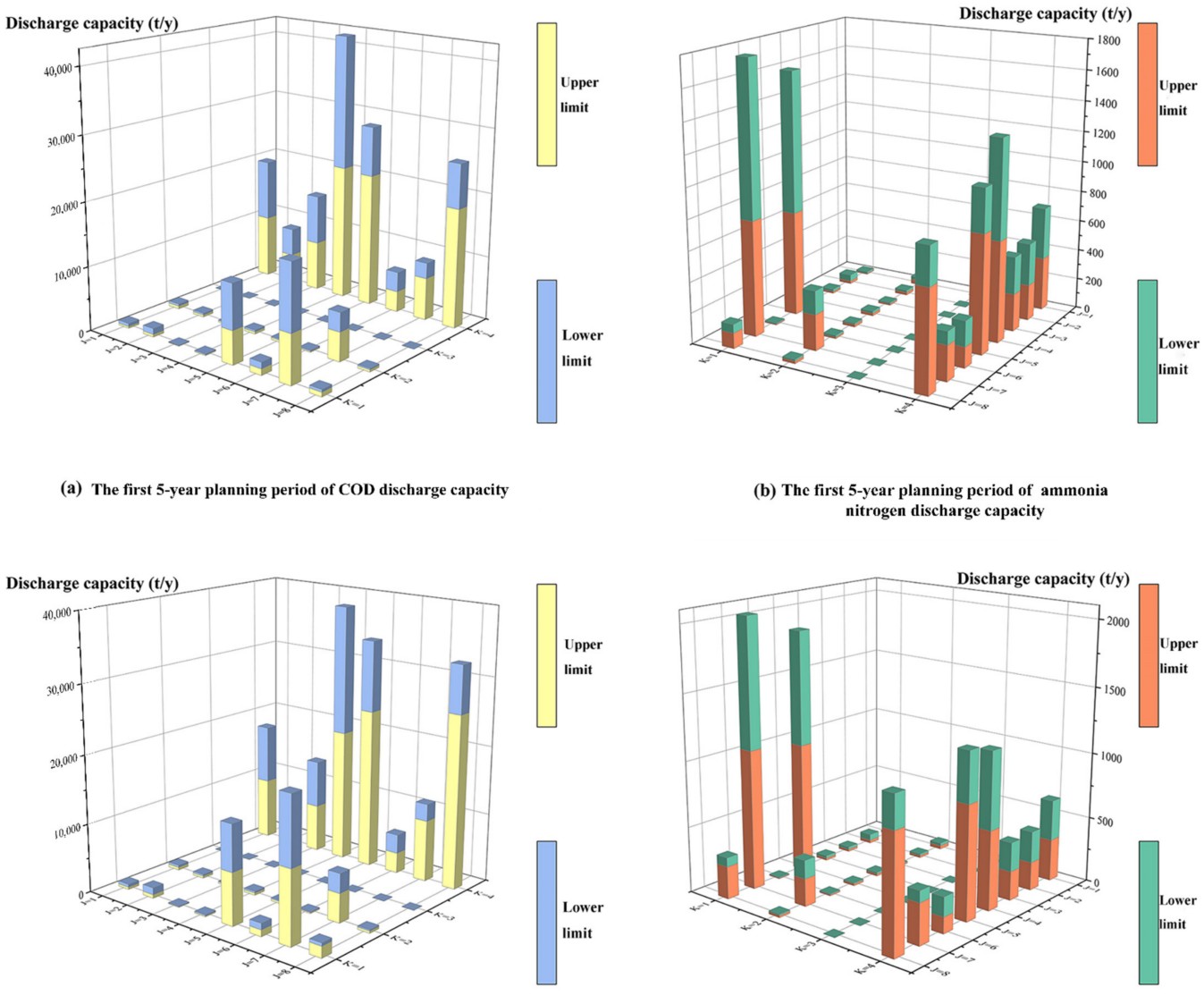

(a) The first 5-year planning period of COD discharge capacity

(b) The first 5-year planning period of ammonia nitrogen discharge capacity

(c) The second 5-year planning period of COD discharge capacity

(d) The second 5-year planning period of ammonia nitrogen discharge capacity

**Figure 2.** *Cont.*

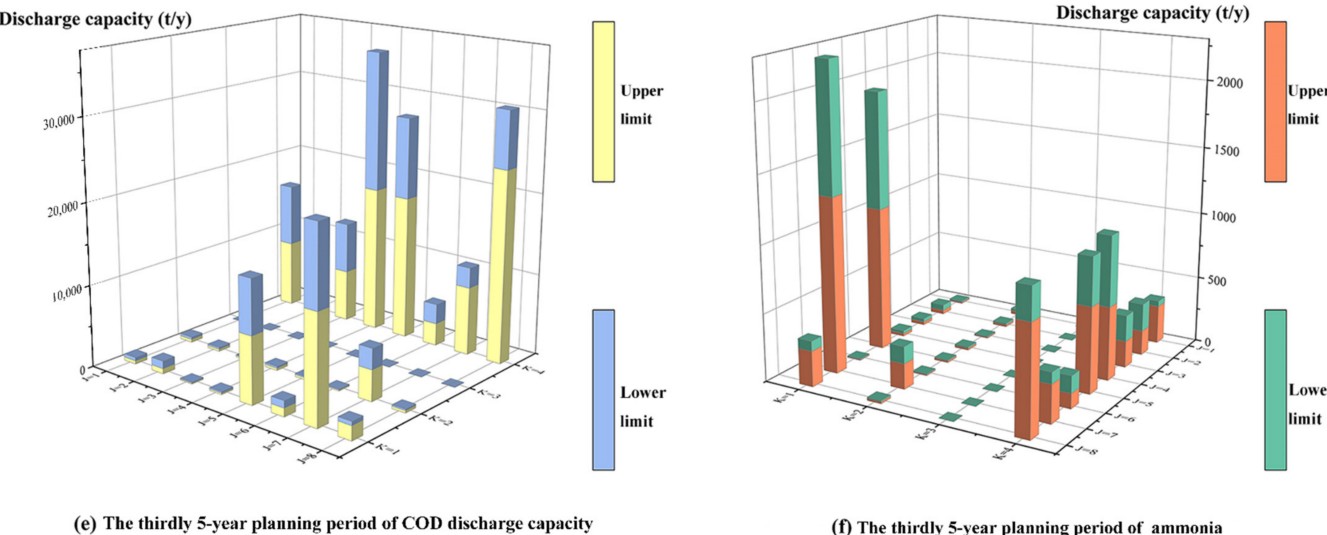

(e) The thirdly 5-year planning period of COD discharge capacity

(f) The thirdly 5-year planning period of ammonia nitrogen discharge capacity

**Figure 2.** COD and ammonia emissions by water-use sectors in the Yinma River Basin for different planning periods. (**a**) The first 5-year planning period of COD discharge capacity by water-use sectors in the Yinma River Basin (**b**) The first 5-year planning period of ammonia emissions discharge capacity by water-use sectors in the Yinma River Basin (**c**) The second 5-year planning period of COD discharge capacity by water-use sectors in the Yinma River Basin (**d**) The second 5-year planning period of second ammonia emissions discharge capacity by water-use sectors in the Yinma River Basin (**e**) The third 5-year planning period of COD discharge capacity by water-use sectors in the Yinma River Basin (**f**) The third 5-year planning period of ammonia emissions discharge capacity by water-use sectors in the Yinma River Basin.

4.2.1. Analysis of Pollutant Discharge Variation in the Yinma River Basin Based on the ITSFR Optimization Method

As an important index to evaluate the water environment of the Yinma River Basin, the amount of pollutant discharge is related to the water ecological security of the Yinma River Basin. Therefore, to further improve the ability of decision makers to control the change of pollutant discharge in the Yinma River Basin and further reduce the risk of water ecological security in the region, this study incorporated a fuzzy optimization method based on the interval-two-stage-robust stochastic programming method into the water resource allocation model of the Yinma River Basin. After the further optimization of the fuzzy optimization method, the upper and lower limits of the discharge amount of the Yinma River Basin were compressed again, which provided a more accurate estimation of the discharge range of the Yinma River Basin for decision makers, who can then adopt or implement more targeted measures and plan according to the overall discharge status of the Yinma River Basin.

As the planning period advanced, the COD emission of industrial sectors in the Yongji area changed from [408.48, 920.06] tonnes/year to [513.15, 1019.01] tonnes/year and then to [640.50, 939.73] tonnes/year. The results showed that the upper limit peak value of the second five-year planning period was higher than that of the other two planning periods, and the lower limit value continued to increase. With the further increase in the planning period, ammonia nitrogen emission increased from [19.09, 43.00] tonnes/year to [23.95, 47.55] tonnes/year and then to [29.89, 43.85] tonnes/year. The results showed that the upper limit peak value of the second five-year planning period was higher than that of the other two planning periods, and the lower limit value continued to increase. In Shuangyang, COD discharge from the industrial sector increased from [56.47, 68.46] tonnes/year to [78.37, 90.28] tonnes/year and then to [106.68, 118.14] tonnes/year with progression of the planning period, and the upper and lower limit values continued to increase. Ammonia nitrogen emission increased from [12.69, 15.38] tonnes/year to [17.35, 19.99] tonnes/year and then to [23.34, 25.84] tonnes/year, and the upper and lower limits of the results continued to increase. In Jiutai, COD discharge from the industrial sector increased from

[112.56, 135.52] tonnes/year to [134.88, 157.67] tonnes/year and then to [164.16, 187.20] tonnes/year, and the upper and lower limits of the results continued to increase. Likewise, the ammonia nitrogen emission increased from [13.94, 16.78] tonnes/year to [16.86, 19.71] tonnes/year and then to [20.11, 22.93] tonnes/year, and the upper and lower limits of the results continued to increase. The results were slightly different for Nong'an compared with those for the other three counties and urban areas; in this region, COD and ammonia nitrogen emissions from industrial sectors varied under different flow availability levels. When the available flow level was low, COD discharge increased from [391.35, 693.84] to [456.04, 1636.68] tonnes/year and then to [485.95, 1747.13] tonnes/year as the planning period advanced, and the upper and lower limits of the results continued to increase. Ammonia nitrogen emission increased from [56.96, 100.98] tonnes/year to [69.22, 248.44] tonnes/year and then to [73.73, 265.07] tonnes/year, and the upper and lower limits of the results continued to increase. When the available flow level was medium, COD discharge increased from [1249.42, 1557.10] tonnes/year to [1636.68, 1636.68] tonnes/year and then to [485.95, 1747.13] tonnes/year. The peak value of the upper limit of the second five-year planning period was higher than that of the other two planning periods, and the lower limit continued to increase. As the planning period advanced, the ammonia nitrogen emission increased from [181.84, 226.62] tonnes/year to [248.43, 248.43] tonnes/year and then to [73.73, 265.07] tonnes/year. The upper and lower limits of the second five-year planning period were higher than those of the other two planning periods. When the available flow level was high, COD discharge increased from [1249.42, 1557.10] tonnes/year to [1636.68, 1636.68] tonnes/year and then to [1704.44, 1747.13] tonnes/year, and the upper and lower limits of the results continued to increase. Ammonia nitrogen emission increased from [181.84, 226.62] tonnes/year to [248.44, 248.44] tonnes/year and then to [258.59, 265.07] tonnes/year, and the upper and lower limits of the results continued to increase.

Under different flow levels available in the eight counties, for different river basin water consumption sectors, different typical emission pollution indices such as COD and ammonia nitrogen varied with the progression of the planning period; the typical emission pollution indices, such as COD and ammonia nitrogen, did not decrease as the planning period advanced in Panshi, Dehui, Yitong, Changchun, Yongji, Shuangyang, Jiutai, and Nong'an. However, in the aforementioned four counties and cities floating was considered only for the industrial water sector. Compared with the results of the original model, the emission ranges of COD and ammonia nitrogen, which include the typical pollution indices of 11 sections in the three planning periods under different available flow levels, were reduced, thereby reflecting the vital role of the fuzzy method; it also shows that the proposed method provides a theoretical basis for decision makers to predict and monitor the water environmental pollution of different regions.

### 4.2.2. Analysis of the Variation of Pollution Indicators into the Yinma River Basin Based on the ITSFR Optimization Method

An analysis of the typical pollutant input provides a clearer and more intuitive understanding of the flow availability of different pollutant types at each section of the Yinma River Basin at low-, medium-, and high-flow levels, as well as the river input during different planning periods. It also provides a clearer and more rational optimization plan for decision makers. Figure 2 depicts the amount of COD and ammonia discharge emissions into the river at each control section in the Yinma River Basin.

The typical pollution indicators, COD and ammonia nitrogen emissions into the river at 11 cross-sections of the Yinma River Basin are shown in Figure 3, and their levels changed as the planning period progressed. Under the low-flow availability scenario, average COD emissions into the 11 river sections during the first five-year planning period were [2442.57, 2844.60] tonnes/year, with the highest COD emissions into the Xinkai River section [(7336.00, 8058.50) tonnes/year]; the average ammonia input to the river was [204.71, 207.69] tonnes/year, and the highest amount of ammonia nitrogen emissions into the river was at the Xinkai River section [(602.13, 619.50) tonnes/year). During the second five-year planning period, the average COD input at the 11 cross-sections

was [2387.15, 3450.88] tonnes/year, with the largest COD input at the Xinkai River section [(7120.00, 10,052.34) tonnes/year]; the average ammonia nitrogen input was [199.30, 270.14] tonnes/year, with the largest ammonia nitrogen input at the same Xinkai River section [(601.88, 788) tonnes/year). The average amount of ammonia nitrogen emissions into the river was [(199.30, 270.14) tonnes/year], and the Xinkai River section had the highest amount of ammonia nitrogen emissions [(601.88, 788.37) tonnes/year]. During the third five-year planning period, the average COD input at the 11 cross-sections was [2299.47, 3471.65] tonnes/year, with the highest COD input at the Xinkai River cross-section [(6904.00, 10,901.97) tonnes/year]; the average ammonia input was [193.02, 281.63] tonnes/year, with the highest ammonia input at the same Xinkai River cross-section [(582.03, 883) tonnes/year]. The average ammonia input to the river was [193.02, 281.63] tonnes/year, with the Xinkai River section having the highest ammonia input [(582.03, 883.20) tonnes/year]. Under the medium-flow availability scenario, the average COD input to the 11 river sections during the first five-year planning period was [2668.92, 3106.84] tonnes/year, with the highest COD emissions into the Xinkai River section [(7758.82, 8554.40) tonnes/year]; the average ammonia input to the river was [230.86, 255.65] tonnes/year, and the amount of ammonia emissions into the Xinkai River section was [230.86, 255.65] tonnes/year. The average amount of ammonia nitrogen entering the river was [230.86, 255.65] tonnes/year, and the same amount of ammonia nitrogen entering the river at the Xinkai River section was the highest [(640.47, 659.13) tonnes/year]. During the second five-year planning period, the average COD input to the 11 river cross-sections was [2613.80, 3450.88] tonnes/year, with the highest COD input to the Xinkai River section [(7503.94, 10,052.34) tonnes/year]; the average ammonia nitrogen emissions into the river was [219.76, 270.14] tonnes/year, with the highest ammonia nitrogen emissions being in the Xinkai River section [(640.86, 788.37) tonnes/year]. During the third five-year planning period, average COD emissions into the 11 river cross-sections were [2401.23, 3471.65] tonnes/year, with the highest COD emissions being in the Xinkai River section [(7249.06, 10901.97) tonnes/year]; the average ammonia nitrogen input to the river was [200.69, 281.63] tonnes/year, with the highest ammonia nitrogen emissions into the Xinkai River section [(616.06, 883.20) tonnes/year]. Under the high-flow availability scenario, the average COD input into the 11 river sections during the first five-year planning period was [2925.85, 3225.83] tonnes/year, with the highest COD emissions into the Xinkai River section [(8202.31, 8895.50) tonnes/year]; average ammonia emissions into the river were [239.36, 241.33] tonnes/year, with the highest ammonia emissions going into the Xinkai River section [(8202.31, 8895.50) tonnes/year]. The average amount of ammonia nitrogen entering the river was [239.36, 241.33] tonnes/year, with the highest amount of ammonia nitrogen entering the river at the Xinkai River section [(658.13, 682.56) tonnes/year]. During the second five-year planning period, the average COD input at the 11 cross-sections was [2851.53, 3450.88] tonnes/year, with the highest COD input at the Xinkai River section [(7906.65, 10,052.34) tonnes/year]; the average ammonia nitrogen input was [232.31, 270.14] tonnes/year, with the highest ammonia nitrogen input at the Xinkai River section [(662.14, 788.14) tonnes/year]. During the third five-year planning period, the average COD input to the 11 river sections was [2567.05, 3471.65] tonnes/year, with the highest COD input to the Xinkai River section [(7610.99, 10,901.97) tonnes/year]; the average ammonia nitrogen input into the river was [219.28, 281.63] tonnes/year, with the highest ammonia nitrogen input into the Xinkai River section [(662.11, 883.20) tonnes/year].

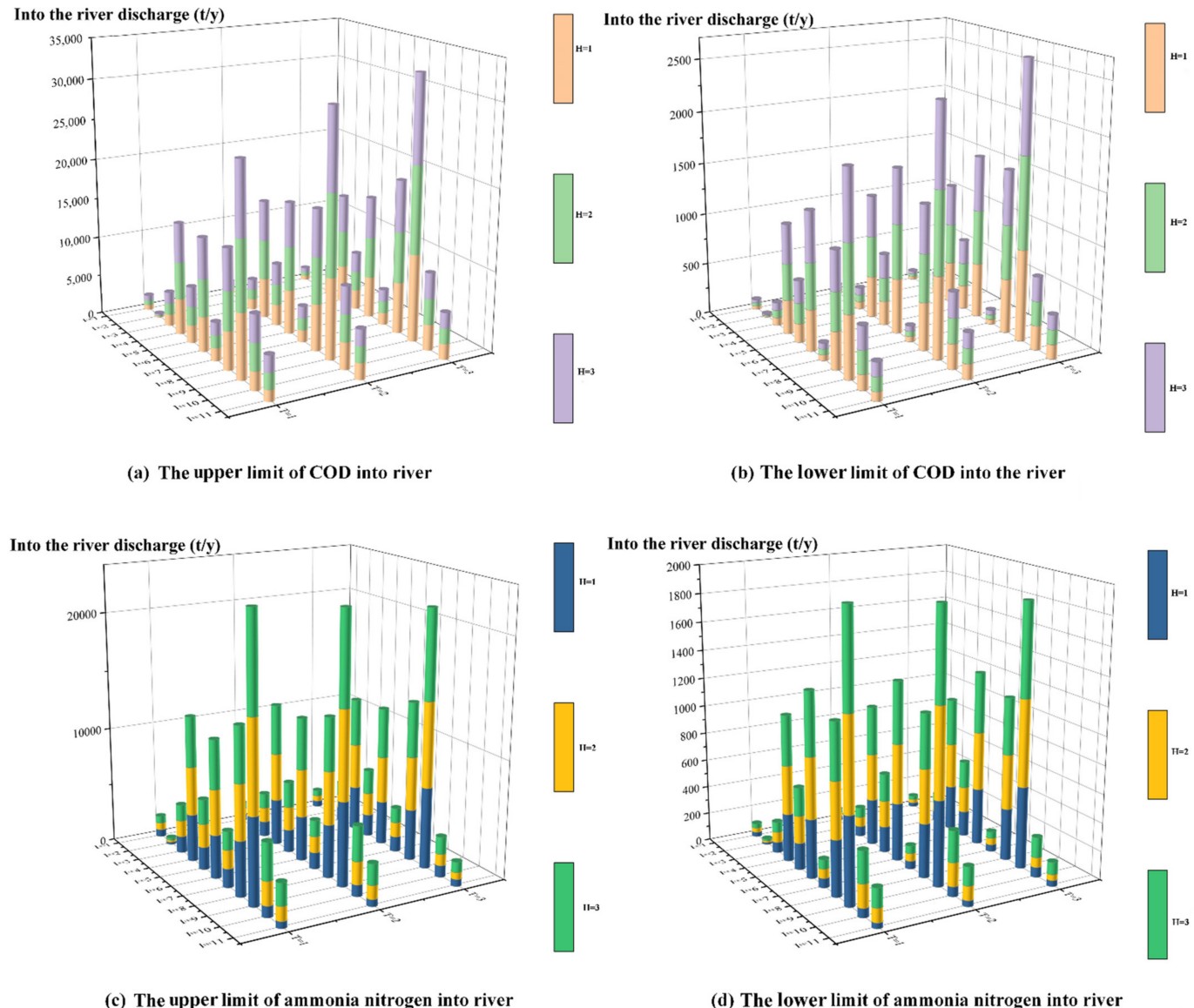

**Figure 3.** COD and ammonia discharge emissions into the river at each control section in the Yinma River Basin. (**a**) The upper limit of COD into river at each control section in the Yinma River Basin (**b**) The lower limit of COD into river at each control section in the Yinma River Basin (**c**) The upper limit of ammonia nitrogen into river at each control section in the Yinma River Basin (**d**) The lower limit of ammonia nitrogen into river at each control section in the Yinma River Basin.

Thus, in the upper reaches of the Yinma River, the Forked River and the Shuangyang River sections, the typical pollution indicators of COD emissions and ammonia nitrogen input into the river did not change with changes in the flow availability level. By contrast, the typical pollution indicators of COD and ammonia nitrogen emissions into the river in the other eight sections changed with changes in the flow availability level; with an increase in the flow availability level from low to high, the typical pollution indicators COD and ammonia nitrogen emissions into the river in the aforementioned eight sections increased. The typical pollution indicators of COD discharge and ammonia nitrogen emissions into the river for the eight cross-sections showed an increasing trend. Similar to the results of the discharge indicator analysis, the typical pollution indicators of COD discharge and ammonia nitrogen emissions at all 11 sections changed as the planning period progressed, that is, the typical pollution indicators of COD discharge and ammonia nitrogen emissions at the eight sections decreased every year with the progression of the planning period. The amounts of COD and ammonia nitrogen emissions into the water environment at the

Xinkai River section were the highest among the 11 sections. More effective treatment measures should be implemented in the area where the Xinkai River section is located to regulate the amount of COD and ammonia nitrogen discharged into the river from various sources in this area, which will help further optimize the water environment of the Yinma River Basin. A comparison of the results with those of the original model showed that the range of discharge intervals for COD and ammonia nitrogen decreased, which further demonstrates the important role of the fuzzy approach.

4.2.3. Analysis of Changes in the Yinma River Basin Capacity Enhancement Project Based on the ITSFR Optimization Method

Due to the local social and economic development needs, as well as the lack of technical advancements, a balance between the discharge volume and the pollutant-carrying capacity of the Yinma River Basin cannot be achieved only by controlling the total number of pollution indicators discharged at the source. Moreover, improving the quality of the already polluted water environment is challenging. To achieve water pollution management in the Yinma River Basin, in addition to considering source control and emission reduction techniques for pollution indicators discharged from outside the river, pollution indicators in the river should be reduced through water pollution control projects and ecological restoration water conservancy projects, which can also improve the pollution-absorption capacity of the water badlands. Among these, the proposed pollution-absorption capacity enhancement project selected artificial wetlands, ecological floating bed, cultured forest, front bank, ecological corridor, artificial aeration, and dredging project.

As shown in Figure 4, in the 11 cross-sections considered in the capacity enhancement project during the first five-year planning period, the average volume of artificial wetlands was [1454.54, 1536.36] tonnes, with a maximum value of [5000.00, 6000.00] tonnes; the average volume of ecological floating beds was [4.09, 13.49] tonnes, with a maximum value of [25.00, 30.00] tonnes; the average volume of culverts was [10.18, 30.00] tonnes, with a maximum value of [80.00, 85.00] tonnes; the average volume of reservoir was [0.64, 1.18] tonnes, with a maximum value of [3.00, 3.00] tonnes; the average volume of ecological corridors was [94.09, 108.18] tonnes, with a maximum value of [250.00, 275.00] tonnes; the average volume of artificial aeration was [27.41, 30.36] tonnes, with a maximum value of [70.00, 80.00] tonnes; and the average volume of dredging works was [75.91, 277.27] tonnes, with a maximum value of [800.00, 1200.00] tonnes. For the second five-year planning period, the corresponding values were [1536.36, 2908.08] tonnes (with a maximum value of [6000.00, 10,000.00]), [4.09, 26.98] (with a maximum value of [25.00, 60.00]), [10.18, 60.00] tonnes (with a maximum value of [85.00, 160.00] tonnes), [0.64, 2.36] tonnes (with a maximum value of [3.00, 6.00] tonnes), [108.18, 188.18] tonnes (with a maximum value of [275, 500] tonnes), [30.36, 54.82] tonnes (with a maximum value of [80.00, 140.00] tonnes), and [75.91, 554.54] tonnes (with a maximum value of [800.00, 2400.00] tonnes). For the third five-year planning period, these values were [1536.36, 4363.62] tonnes (with a maximum value of [6000.00, 15,000.00] tonnes), [4.09, 40.47] tonnes (with a maximum value of [25.00, 90.00] tonnes), [10.18, 90.00] tonnes (with a maximum value of [85.00, 240.00] tonnes), [0.64, 3.54] tonnes (with a maximum value of [3.00, 9.00] tonnes), [108.18, 282.27] tonnes (with a maximum value of [275.00, 750.00] tonnes), [30.36, 82.23] tonnes (with a maximum value of [80.00, 210.00] tonnes), and [75.91, 831.11] tonnes (with a maximum value of [800.00, 3600.00] tonnes), respectively.

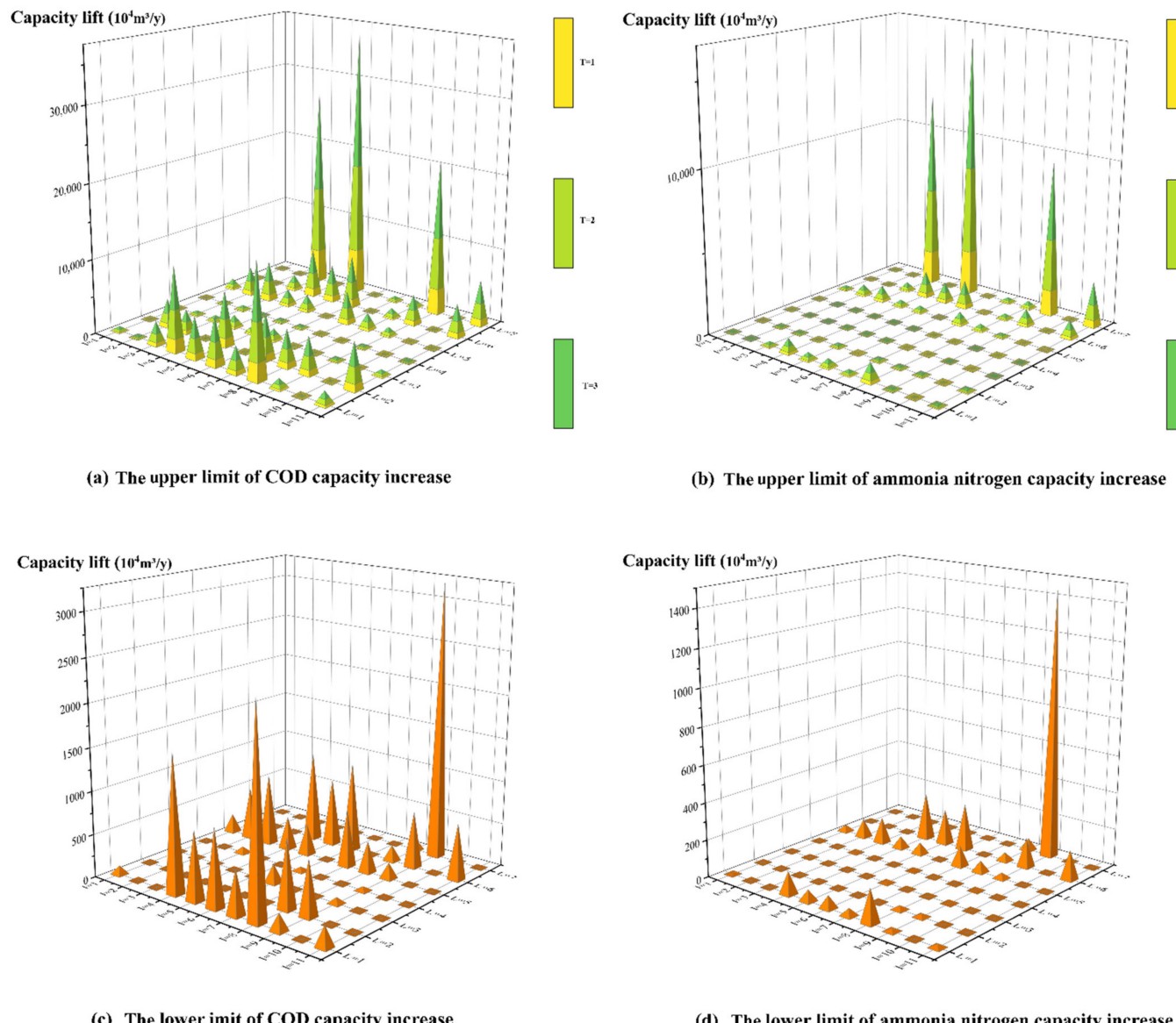

**Figure 4.** COD and ammonia nitrogen capacity increase at each control section in the Yinma River Basin.(**a**) The upper limit of COD capacity increase at each control section in the Yinma River Basin (**b**) The upper limit of ammonia nitrogen capacity increase at each control section in the Yinma River Basin (**c**) The lower limit of COD capacity increase at each control section in the Yinma River Basin (**d**) The lower limit of ammonia nitrogen capacity increase at each control section in the Yinma River Basin.

In the three planning periods, the 11 cross-sections, with the exception of the midstream cross-section of the Yitong River, will require maximum work on artificial wetland enhancement; the other 10 cross-sections will require artificial wetland enhancement works to enhance the pollution-absorption capacity of the water environment in the Yinma River Basin. During the three planning periods, among the 11 cross-sections, the Shuangyang, Wukai, and Xinkai River cross-sections should adopt all the aforementioned seven types of pollution-absorption functions, including artificial wetlands, to improve the pollution-absorption capacity of the water environment in the Yinma River Basin. The middle reaches of the Yitong River need not be upgraded to improve the pollution-absorption capacity of the Yinma River Basin for this cross-section. By comparing the results with those of the original model, the capacity enhancement project for the 11 cross-sections has been upgraded from the first five-year planning period only to the optimization of the capacity enhancement project for the Yinma River water environment in the three planning periods

by means of fuzzy optimization, which further refines the optimization results to help decision makers in providing better solutions for the overall pollution and prevention of the Yinma River water environment.

4.2.4. Analysis of Capacity Improvement of Pollution Indicators in the Yinma River Basin Based on the ITSFR Optimization Method

Increasing efforts have been directed towards mitigating water pollution in the form of preventative measures in various regions, along with the implementation of water ecological restoration projects. However, the relationship between the total number of river pollution indicators and the limit of the approved pollutant-carrying capacity cannot accurately reflect the trend in water environmental quality. Therefore, this study aims to build an index system for improving the water-pollutant-carrying capacity of the Yinma River Basin, realize the organic combination of 'total source reduction' and 'sink treatment improvement' in water environment pollution control and provide a basis for further research on the total emission control and water pollution management of the Yinma River Basin. Furthermore, the study aims to determine the total pollution index control of the Yinma River Basin based on the improvement of its pollutant-carrying capacity.

This study also optimizes the sewage-carrying capacity for each section of the Yinma River Basin. Based on the sewage-carrying capacity improvement project, the typical pollution indicators—COD and ammonia nitrogen capacity—of each section of the Yinma River Basin exhibited a potential for optimization and improvement. During the initial five-year planning period, the constructed wetland project was adopted for each section. Herein, the COD capacity of each section was increased by an average of [614.55, 727.27], with the maximum capacity being observed in the Changchun City section of the Yitong River ([2400.00, 2500.00] tonnes). Moreover, the ammonia nitrogen capacity witnessed an average increase of [46.09, 54.55] tonnes, with the maximum value observed in the Changchun City section of the Yitong River ([180.00, 187.50] tonnes). The ecological floating bed project was adopted for each section to increase the COD capacity of each section by an average of [130.91, 536.36] tonnes, with the maximum value observed in the Changchun section of Wukai and Yitong Rivers [(800.00, 1200.00) tonnes]. Meanwhile, the ammonia nitrogen capacity increased by an average of [3.44, 14.08] tonnes, with the maximum increase noted in the Changchun section of Yitong and Wukai Rivers [(21.00, 31.50) tonnes]. Furthermore, the conservation forest project adopted for each section increased the COD capacity of each section by an average of [26.47, 97.50] tonnes, with the maximum value observed in the lower and middle reaches of the Yinma River [(221.00, 260.00) tonnes]. The ammonia nitrogen capacity increased by an average of [0.82, 3.00] tonnes, with the maximum value noted in the lower and middle reaches of the Yinma River [(6.80, 8.00) tonnes]. The forebay project adopted for each section led to an average increase in the COD capacity of each section by [14.89, 34.57] tonnes, with a maximum increase noted in the section of the Shuangyang River [(70.20, 87.75) tonnes]. The ammonia nitrogen capacity increased by an average of [0.99, 2.29] tonnes, whereas the maximum increase was noted for the Shuangyang River section [(4.65, 5.81) tonnes). The ecological corridor project was adopted for each section to increase the COD capacity of each section by an average of [324.55, 329.32] tonnes, with the maximum increase noted in the Shuangyang River section [(825.00, 875.00) tonnes]. The ammonia nitrogen capacity increased by an average of [54.09, 56.46] tonnes, with the maximum increase noted in the Shuangyang River section [(137.50, 150.00) tonnes]. Artificial aeration works were adopted for each section to increase the COD capacity of each section by an average of [394.73, 445.40] tonnes, with the maximum value noted for the middle reaches and the downstream sections of the Yinma River [(1040.00, 1137.50) tonnes]. The ammonia nitrogen capacity increased by an average of [95.34, 107.58] tonnes, with the maximum value noted in the downstream sections of the Yinma River [(251.20, 274.75) tonnes]. Desilting works were adopted for each section to increase the COD capacity of each section by an average of [303.64, 1386.36] tonnes, with the maximum increase observed in the sections of Xinkai and Wukai Rivers [(3200.00, 6000.00) tonnes]. The ammonia nitrogen capacity increased by an average of [136.64, 623.86] tonnes,

with the maximum increase observed in sections of Xinkai and Wukai Rivers [(1440.00, 2700.00) tonnes].

During the second five-year planning period, the constructed wetland project was adopted for each section to increase the COD capacity of each section by an average of [614.55, 1454.55] tonnes, whereas the maximum increase was noted in the Changchun City section of the Yitong River [(2400.005000.00) tonnes]. The ammonia nitrogen capacity increased by an average of [46.09, 109.09] tonnes, with the maximum increase noted in the Changchun City section of the Yitong River [(180.00, 375.00) tonnes]. The ecological floating bed project was adopted for each section to increase the COD capacity of each section by an average of [130.91, 1072.73] tonnes, whereas the maximum value was noted for the Changchun section of the Yitong River and a section of the Wukai River [(800.00, 2400.00) tonnes]. The ammonia nitrogen capacity increased by an average of [3.44, 28.16] tonnes, whereas the maximum increase was observed in the Changchun section of the Yitong River and the section of the Wukai River ([21.00, 63.00] tonnes). The conservation forest project was adopted for each section to increase the COD capacity of each section by an average of [25.47, 195.00] tonnes, with the maximum increase noted for the lower and middle reaches of the Yinma River [(221.00, 520.00) tonnes]. The ammonia nitrogen capacity increased by an average of [0.82, 6.00] tonnes, whereas the maximum increase was noted for the lower and middle reaches of the Yinma River [(6.80, 16.00) tonnes]. The forebay project was adopted to increase the COD capacity of each section by an average of [14.89, 69.14] tonnes, with the maximum value noted for the Shuangyang River section [(70.20, 175.50) tonnes]. The ammonia nitrogen capacity increased by an average of [0.99, 4.58] tonnes, whereas the maximum increase was noted in the Shuangyang River section [(4.65, 11.63) tonnes]. The ecological corridor project was adopted to increase the COD capacity of each section by an average of [324.55, 658.64] tonnes, with the maximum value noted in the Shuangyang River section [(825.00, 1750.00) tonnes]. The ammonia nitrogen capacity increased by an average of [54.09, 112.91] tonnes, with the maximum value noted in the Shuangyang River section [(137.50, 300.00) tonnes]. The artificial aeration project was adopted to increase the COD capacity of each section by an average of [394.73, 890.80] tonnes, with the maximum value observed in the middle reaches and the downstream section of the Yinma River [(1040.00, 2275.00) tonnes]. The ammonia nitrogen capacity increased by an average of [95.34, 215.16] tonnes, whereas the maximum increase was noted for the downstream section of the Yinma River ([251.20, 549.50] tonnes). Desilting works were adopted to increase the COD capacity of each section by an average of [303.64, 2772.73] tonnes, with the maximum value observed for the Xinkai and Wukai River sections [(3200.00, 12,000.00) tonnes]. The ammonia nitrogen capacity increased by an average of [136.64, 1247.73] tonnes, with the maximum value observed for the sections of Xinkai and Wukai Rivers [(1440.00, 5400.00) tonnes].

During the third five-year planning period, the constructed wetland project was adopted for each section to increase the COD capacity of each section by an average of [6144.55, 2181.82] tonnes, with the maximum value noted for the Changchun City section of the Yitong River [(2400.00, 7500.00) tonnes]. The ammonia nitrogen capacity increased by an average of [46.09, 163.64] tonnes, with the maximum increase noted for the Changchun City section of the Yitong River [(180.00, 562.50) tonnes]. The ecological floating bed project was adopted for each section to increase the COD capacity of each section by an average of [130.91, 1609.09] tonnes, with the maximum increase noted for the Changchun section of the Yitong River and the Wukai River section [(800.00, 3600.00) tonnes]. The ammonia nitrogen capacity increased by an average of [3.44, 42.24] tonnes, whereas the maximum increase was noted for the Changchun section of the Yitong and Wukai River sections [(21.00, 94.50) tonnes]. The conservation forest project was adopted to increase the average COD capacity of each section by an average of [25.47, 292.50] tonnes, with the maximum increase noted in the downstream section of the Yinma River [(221.00, 780.00) tonnes]. The ammonia nitrogen capacity increased by an average of [0.82, 9.00] tonnes, with the maximum increase noted in the downstream sections and middle reaches of the Yinma River [(6.80, 24.00) tonnes]. The

forebay project was adopted for each section, which resulted in an average increase in the COD capacity of each section by [14.89, 103.71] tonnes, with the maximum increase noted in the section of the Shuangyang River [(70.20, 263.25) tonnes]. The ammonia nitrogen capacity was increased by an average of [0.99, 6.87] tonnes, with the maximum value observed in the Shuangyang River section [(4.65, 17.45) tonnes]. The ecological corridor project was adopted to increase the COD capacity of each section by an average of [324.55, 987.96] tonnes, with the maximum increase noted for the Shuangyang River section [(825.00, 2625.00) tonnes]. The ammonia nitrogen capacity increased by an average of [54.09, 169.37] tonnes, with the maximum increase observed for the Shuangyang River section [(137.50, 450.00) tonnes]. The artificial aeration project was adopted to increase the COD capacity of each section by an average of [394.73, 1336.20] tonnes, with the maximum increase noted for the middle reaches and the downstream section of the Yinma River [(1040.00, 3412.50) tonnes]. The ammonia nitrogen capacity increased by an average of [95.34, 322.74] tonnes, whereas the maximum value was observed in the downstream section of the Yinma River [(251.20, 824.25) tonnes]. Desilting works were adopted to increase the COD capacity of each section by an average of [303.64, 4159.10] tonnes, with the maximum value noted for the Xinkai and Wukai River sections [(3200.00, 18,000.00) tonnes]. The ammonia nitrogen capacity increased by an average of [136.64, 1871.60] tonnes, with the maximum value noted in Xinkai and Wukai River sections [(1400.00, 8100.00) tonnes].

It is evident from Figure 4 that in the three planning periods, except for the typical pollutants (COD and ammonia nitrogen) of the water environment in the middle reaches of the Yitong River, the environmental capacity of carrying the two typical pollution indicators in the other sections was improved. Among them, through the sewage-carrying capacity improvement project, the total increase in the COD and ammonia nitrogen of typical pollutants in the water environment of the Wukai River section was [2248.40, 12,581.13] tonnes, [2248.40, 25,162.25] tonnes, and [2248.40, 37,743.38] tonnes, thus ranking first in causing an increase in the watershed capacity of typical pollutants in the water environment of each section. The aforementioned results indicate that there is an inflow of effluents from multiple upstream sewage treatment plants to the middle reaches of Yitong River. Consequently, the water environment pollutant-carrying pressure of the watershed in this area is large. Even though the application of the proposed basin pollutant-carrying capacity improvement project, the goal of improving the typical pollutant capacity of the modified section watershed is unachievable. However, through the sewage-carrying capacity improvement project, there is a marked overall capacity improvement in other sections for typical pollutants (COD and ammonia nitrogen) in the water environment. This can alleviate the water environment pressure in the section in the middle reaches of the Yitong River. Compared with results of the original model, with the help of the fuzzy optimization method, the pollutant-carrying capacity of the 11 sections exhibited a marked improvement from optimization and upgradation of the first five-year planning period to the optimization of the pollutant capacity improvement of the Yinma River environment in the three planning periods. Therefore, this study optimized the capacity improvement to help decision makers propose a more efficient scheme for the prevention and control of the water pollution in the Yinma River Basin.

*4.3. Analysis of Changes in the Water Allocation Scheme Based on the ITSFR Optimization Method*
4.3.1. Analysis of Changes in Water Allocation Based on the ITSFR Optimization Method

The aim is to further reduce the decision space and to enable decision makers to more accurately and appropriately adjust water-use quotas for various water sectors in the cities and counties of the Yinma Basin to accommodate the adjustments that will be made as the planning period progresses. This study introduces fuzzy optimization based on interval two-stage robust (ITSR) optimization methods. The aim is to employ the fuzzy optimization methods to efficiently allocate available water resources in the Yinma River Basin to each water-use unit in the planning area in a uniform manner, ultimately resulting in economic and ecological benefits. Table 2 presents the allocation of water resources to

different water-use sectors in various planning areas in the Yinma River Basin for each planning period.

**Table 2.** Water allocation for different water-use sectors in various planning areas in the Yinma River Basin for each planning period ($10^4$ m$^3$/year).

| Sectors | h = 1 | h = 2 | h = 3 |
|---|---|---|---|
| Industrial | [573.60, 758.00] | [768.80, 1089.00] | [1047.37, 1047.37] |
| Municipal | [1380.00, 1736.00] | [1387.20, 1754.00] | [1393.60, 1771.00] |
| Ecological | [396.00, 498.00] | [435.00, 572.40] | [479.00, 657.60] |
| Agricultural | [20,727.00, 21,945.00] | [20,748.00, 22,250.00] | [21,301.00, 23,461.00] |
| Industrial | [222.00, 222.00] | [311.00, 311.00] | [287.20, 427.00] |
| Municipal | [1110.97, 1341.00] | [1062.40, 1362.00] | [1066.40, 1332.59] |
| Ecological | [304.00, 382.80] | [334.00, 439.20] | [368.00, 505.20] |
| Agricultural | [8939.00, 9465.00] | [8949.00, 9597.00] | [9187.00, 9609.49] |
| Industrial | [772.00, 772.00] | [1244.00, 1244.00] | [1961.00, 1961.00] |
| Municipal | [641.60, 815.00] | [649.60, 836.00] | [658.40, 857.00] |
| Ecological | [182.00, 229.20] | [200.00, 264.00] | [221.00, 303.60] |
| Agricultural | [9116.00, 9487.00] | [9074.00, 9481.00] | [9430.00, 9980.00] |
| Industrial | [1738.09, 1806.00] | [2551.82, 2654.00] | [3651.80, 3800.00] |
| Municipal | [1121.60, 1416.00] | [1130.40, 1437.00] | [1138.40, 1459.00] |
| Ecological | [326.00, 410.40] | [366.00, 480.00] | [410.00, 561.60] |
| Agricultural | [26,298.00, 27,523.00] | [26,431.00, 27,905.00] | [27,734.00, 29,801.00] |
| Industrial | [4475.00, 4732.00] | [6144.00, 6144.00] | [8432.00, 8432.00] |
| Municipal | [1026.40, 1282.61] | [1032.00, 1289.61] | [1036.80, 1295.60] |
| Ecological | [300.00, 376.80] | [336.00, 440.40] | [376.00, 516.00] |
| Agricultural | [45,819.00, 45,819.00] | [46,051.00, 46,051.00] | [48,321.00, 48,321.00] |
| Industrial | [538.00, 538.00] | [610.00, 610.00] | [689.00, 689.00] |
| Municipal | [553.00, 641.45] | [440.80, 559.00] | [442.40, 564.00] |
| Ecological | [126.00, 157.20] | [138.00, 180.00] | [152.00, 204.00] |
| Agricultural | [12,696.00, 13,536.00] | [12,566.00, 13,921.00] | [12,792.00, 14,725.00] |
| Industrial | [11,165.68, 12,033.00] | [17,995.92, 18,062.16] | [24,922.75, 28,075.39] |
| Municipal | [16,685.10, 18,533.34] | [16,894.80, 19,295.00] | [17,107.20, 19,782.00] |
| Ecological | [5081.05, 5523.60] | [5072.00, 6628.80] | [5833.00, 7953.60] |
| Agricultural | [11,272.00, 11,272.00] | [11,329.00, 11,961.00] | [11,887.00, 12,774.00] |
| Industrial | [1035.90, 1291.00] | [1680.37, 1727.00] | [2200.93, 2269.00] |
| Municipal | [1109.60, 1386.58] | [1128.80, 1451.00] | [1148.80, 1495.00] |
| Ecological | [319.00, 400.80] | [357.00, 469.20] | [400.00, 548.40] |
| Agricultural | [61,958.00, 61,958.00] | [62,271.00, 65,744.00] | [65,342.00, 70,211.00] |

As shown in Table 2, the average water allocation for the industrial sector in the eight planning areas during the first five-year planning period was [2565.03, 2769.00] $\times$ $10^4$ m$^3$/year, with the highest industrial sector water allocation observed in Changchun [11,165.68, 12,033.00] $\times$ $10^4$ m$^3$/year; meanwhile, the average water allocation for the municipal sector was [2964.59, 3382.94] $\times$ $10^4$ m$^3$/year, with the highest amount of water allocation for the municipal sector occurring in Changchun [16,685.10, 18,533.34] $\times$ $10^4$ m$^3$/year; the average water allocation for the ecological sector was [879.26, 997.35] $\times$ $10^4$ m$^3$/year, with the highest value of water allocation for the ecological sector occurring in Changchun [5081.05, 5523.60] $\times$ $10^4$ m$^3$/year; the average water allocation in the agricultural sector was [24,603.13, 25,125.63] $\times$ $10^4$ m$^3$/year, with the highest amount of water allocation in the agricultural sector observed in Nong'an [61,958.00, 61,958.00] $\times$ $10^4$ m$^3$/year. The average water allocation for the industrial sector in the eight planning areas for the second five-year planning period was [3921.52, 3971.87] $\times$ $10^4$ m$^3$/year, with the highest industrial sector water allocation noted for Changchun [17,995.92, 18,062.16] $\times$ $10^4$ m$^3$/year; the average water allocation for the municipal sector was [2965.75, 3497.95] $\times$ $10^4$ m$^3$/year, with the highest value of water allocation for the municipal sector occurring

in Changchun [16,894.80, 19,295.00] $\times$ $10^4$ m$^3$/year. Furthermore, the average water allocation for the ecological sector was [904.75, 1184.25] $\times$ $10^4$ m$^3$/year, with the highest amount of water allocation for the ecological sector occurring in Changchun [5072.00, 6628.80] $\times$ $10^4$ m$^3$/year; the average water allocation in the agricultural sector was [24,677.38, 25,863.75] $\times$ $10^4$ m$^3$/year, with the highest amount of water allocation in the agricultural sector occurring in Nong'an [62,271.00, 65,744.00] $\times$ $10^4$ m$^3$/year. Moreover, the average water allocation for the industrial sector in the eight planning areas for the third five-year planning period was [5426.04, 5810.56] $\times$ $10^4$ m$^3$/year, with the highest industrial sector water allocation occurring in Changchun [24,922.75, 28,075.39] $\times$ $10^4$ m$^3$/year; the average water allocation for the municipal sector was [2999.00, 3519.53] $\times$ $10^4$ m$^3$/year, with the highest amount of water allocation for the municipal sector observed for Changchun [17,107.20, 19,782.00] $\times$ $10^4$ m$^3$/year. The average water allocation for the ecological sector was [1029.88, 1406.25] $\times$ $10^4$ m$^3$/year, with the highest amount of water allocation for the ecological sector occurring in Changchun [5833.00, 7953.60] $\times$ $10^4$ m$^3$/year; the average water allocation in the agricultural sector was [25,749.25, 27,360.31] $\times$ $10^4$ m$^3$/year, with the highest amount of water allocation noted for Nong'an [65,342.00, 74,0211.00] $\times$ $10^4$ m$^3$/year.

The aforementioned results clearly indicate that Changchun has the highest industrial, municipal, and ecological water use compared to the other seven planning areas over the three planning periods, and therefore, the region has a superior economic output. The overall allocation of water resources in the Yinma River Basin has decreased, indicating that after adjustment, the demand for water resource allocation in the planning area has further reduced. The result implies that normal water demand in the region can be met through the allocation of fewer water resources. After adjustment, the allocation of fewer water resources can satisfy the needs of different water-use sectors in different planning areas in each period. Moreover, after adjustment, the proportion of water resources allocated to each region did not change significantly. With the exception of Changchun, the production water use in other planning areas is still dominated by the agricultural sector, accounting for more than 80% of the allocated regional water resources. After optimization of allocation of water resources by the fuzzy method, the water requirement of the Yinma River Basin is further reduced compared with that achieved using the ITSR stochastic programming method. It further provides an effective framework for decision makers to optimize the allocation of water resources of the four major water-use sectors in eight counties, cities, and districts in the Yinma River Basin.

4.3.2. Analysis of Water Deficit Variation by Water-Use Sector in Each Planning Area Based on the ITSFR Optimization Method

The greater the deficit, the greater is the gap between the actual water supply and demand, and the less stable is the water supply. With implementation of the fuzzy optimization method, the deficit in the Yinma River Basin water resource allocation tends to shrink, demonstrating the functionality of the fuzzy optimization method. The water deficits for each water-use sector in different planning areas, various water-use sectors, different planning periods, and different levels of flow availability are shown in Figure 5.

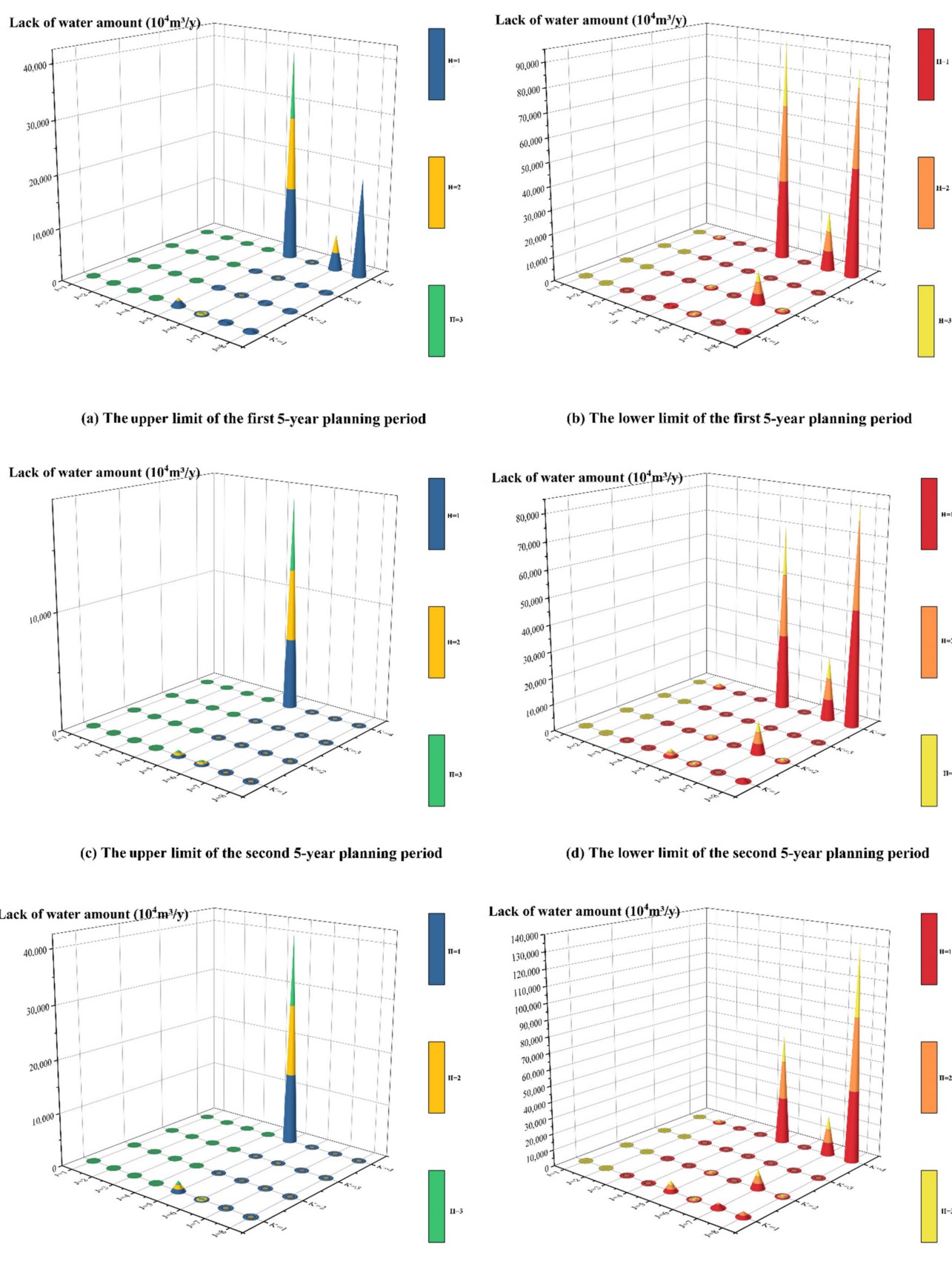

**Figure 5.** Water deficits by water-use sector ($10^4$ m$^3$/year) under different scenarios for various planning periods in the Yinma River Basin. (**a**) The upper limit of the first 5-year planning period water deficits by water-use sector under different

scenarios for various planning periods in the Yinma River Basin (**b**) The lower limit of the first 5-year planning period water deficits by water-use sector under different scenarios for various planning periods in the Yinma River Basin (**c**) The upper limit of the second 5-year planning period water deficits by water-use sector under different scenarios for various planning periods in the Yinma River Basin (**d**) The lower limit of the second 5-year planning period water deficits by water-use sector under different scenarios for various planning periods in the Yinma River Basin (**e**) The upper limit of the third 5-year planning period water deficits by water-use sector under different scenarios for various planning periods in the Yinma River Basin (**f**) The lower limit of the third 5-year planning period water deficits by water-use sector under different scenarios for various planning periods in the Yinma River Basin.

In this optimization study on the four major water-use sectors, Panshui, Shuangyang, and Jiutai have a water deficit of $0 \times 10^4$ m$^3$/year under different flow availability level scenarios and planning periods (Figure 5). In Yongji, under different flow availability level scenarios and planning periods, the industrial, municipal, and ecological sector water deficit was $0 \times 10^4$ m$^3$/year. The agricultural sector water deficit is not affected by the flow availability level; the first, second, and third five-year planning period water deficits were [0.00, 244.69] $\times 10^4$ m$^3$/year, [0.00, 667.59] $\times 10^4$ m$^3$/year, and [0.00, 676.59] $\times 10^4$ m$^3$/year, respectively. In Dehui, the shortfall for the first five-year planning period under the low-flow availability level scenario for industrial water was [120.89, 1057.77] $\times 10^4$ m$^3$/year, whereas the shortfall for the second five-year planning period was [163.22, 854.16] $\times 10^4$ m$^3$/year, and that for the third five-year planning period was [656.98, 2128.84] $\times 10^4$ m$^3$/year. Furthermore, the shortfall for the medium-flow availability level scenario was [0.00, 479.52] $\times 10^4$ m$^3$/year for the first five-year planning period and [163.22, 874.54] $\times 10^4$ m$^3$/year for the second five-year planning period. The shortfall for the first five-year planning period was [0.00, 479.52] $\times 10^4$ m$^3$/year, that for the second five-year planning period was [163.22, 874.54] $\times 10^4$ m$^3$/year, and that for the third five-year planning period was [656.98, 2154.78] $\times 10^4$ m$^3$/year under the medium-flow availability level scenario. Similarly, the shortfall under the high-flow availability level scenario was [0.00, 479.52] $\times 10^4$ m$^3$/year. The water deficit for the first five-year planning period was [0.00, 0.00] $\times 10^4$ m$^3$/year, that for the second five-year planning period was [163.22, 895.91] $\times 10^4$ m$^3$/year, and that for the third five-year planning period was [656.98, 2181.99] $\times 10^4$ m$^3$/year. Municipal water deficits are unaffected by the level of flow availability and were found to be [0.00, 329.66] $\times 10^4$ m$^3$/year, [0.00, 322.11] $\times 10^4$ m$^3$/year, and [0.00, 279.70] $\times 10^4$ m$^3$/year for the first, second, and third five-year planning periods, respectively. The water deficit in the ecological water-use sector was $0 \times 10^4$ m$^3$/year. The agricultural sector remains a major water user and water deficit user, with a deficit of [13,755.03, 34,183.91] $\times 10^4$ m$^3$/year for the first five-year planning period, that of [6062.50, 26,883.97] $\times 10^4$ m$^3$/year for the second five-year planning period, and that of [13,531.62, 29,119.92] $\times 10^4$ m$^3$/year for the third five-year planning period under the low-flow availability level scenario. Moreover, the deficit was observed to be [13,755.02, 32,463.92] $\times 10^4$ m$^3$/year for the first five-year planning period under the medium-flow availability level scenario, whereas the water deficits for the second and third five-year planning periods under this scenario were [6062.50, 23,700.35] $\times 10^4$ m$^3$/year and [13,531.62, 23,683.82] $\times 10^4$ m$^3$/year, respectively. Under the high-flow availability level scenario, the water deficit for the first five-year planning period was [12,541.93, 27,885.83] $\times 10^4$ m$^3$/year, that for the second five-year planning period was [6062.50, 18,494.07] $\times 10^4$ m$^3$/year, and that for the third five-year planning period was [13,531.62, 17,981.96] $\times 10^4$ m$^3$/year. In Yitong, under different flow availability level scenarios and different planning periods, the municipal, ecological and agricultural sectors exhibited a water deficit of $0.00 \times 10^4$ m$^3$/year in this optimization study. The water deficit of the industrial sector does not depend on the flow availability level, with a water deficit of [94.94, 94.94] $\times 10^4$ m$^3$/year for the first five-year planning period, whereas those of [108.71, 131.5] $\times 10^4$ m$^3$/year and [41.19, 147.93] $\times 10^4$ m$^3$/year for the second and third five-year planning periods, respectively. In Changchun, under the low-flow availability level scenario for industrial water use, the deficits for the first, second, and third five-year planning periods were [0.00, 260.86] $\times 10^4$ m$^3$/year, $0.00 \times 10^4$ m$^3$/year,

and [0.00, 2451.58] $\times 10^4$ m$^3$/year, respectively. Under the medium-flow availability level scenario, the deficit was [0.00, 799.45] $\times 10^4$ m$^3$/year, except for the third five-year planning 0.00 $\times 10^4$ m$^3$/year for the planning period, except for the third five-year planning period, where the shortfall is [0.00, 799.45] $\times 10^4$ m$^3$/year. For the high-flow availability level, the shortfall was 0.00 $\times 10^4$ m$^3$/year for all three planning periods. For municipal water, under the three scenarios of flow availability levels (low, medium, and high), the water deficits for the first, second, and third five-year planning periods were [0.00, 4512.68] $\times 10^4$ m$^3$/year, [0, 4021.53] $\times 10^4$ m$^3$/year, and [0.00, 3936.70] $\times 10^4$ m$^3$/year, respectively, with an ecological water deficit of 0.00 $\times 10^4$ m$^3$/year. For the agricultural sector, water deficits of [3383.89, 8409.63] $\times 10^4$ m$^3$/year, [0.00, 8093.24] $\times 10^4$ m$^3$/year, and [0.00, 8173.59] $\times 10^4$ m$^3$/year were noted for the first, second, and third five-year planning periods under the low-flow availability level scenario. The values under the medium-flow availability level were the same as those under the low-flow availability level scenario; the high-flow availability level scenario for the agricultural sector exhibited reduced deficits of [0.00, 8409.63] $\times 10^4$ m$^3$/year, [0.00, 8093.24] $\times 10^4$ m$^3$/year, and [0.00, 8173.59] $\times 10^4$ m$^3$/year for the first, second, and third five-year planning periods, respectively. For Nong'an, the low-flow availability level scenario for the industrial sector exhibited a deficit of [460.63, 966.53] $\times 10^4$ m$^3$/year for the first five-year planning period, that of [0.00, 1245.79] $\times 10^4$ m$^3$/year for the second five-year planning period, and that of [0.00, 1637.90] $\times 10^4$ m$^3$/year for the third five-year planning period. Furthermore, the medium-flow availability level scenario exhibited deficits of 0.00 $\times 10^4$ m$^3$/year, 0.00 $\times 10^4$ m$^3$/year, and [0.00, 1637.90] $\times 10^4$ m$^3$/year for the first, second, and third five-year planning periods, respectively. Under the medium-flow availability scenario, the shortfall was 0.00 $\times 10^4$ m$^3$/year for the first five-year planning period, 0.00 $\times 10^4$ m$^3$/year for the second five-year planning period, and [0.00, 1637.90] $\times 10^4$ m$^3$/year for the third five-year planning period. Under the high-flow availability scenario, the shortfall was 0.00 $\times 10^4$ m$^3$/year for the first five-year planning period and [0.00, 1637.90] $\times 10^4$ m$^3$/year for the second five-year planning period. The second five-year planning period deficit was 0.00 $\times 10^4$ m$^3$/year, whereas deficit in the third five-year planning period was reduced to [0.00, 55.45] $\times 10^4$ m$^3$/year. The water deficits for the municipal sector under the low-, medium-, and high-flow availability level scenarios were [0.00, 324.17] $\times 10^4$ m$^3$/year for the entirety of the first five-year planning period, [0.00, 312.19] $\times 10^4$ m$^3$/year for the entirety of the second five-year planning period, and [0.00, 253.74] $\times 10^4$ m$^3$/year for the entirety of the third five-year planning period. The ecological sector water exhibited a deficit of 0.00 $\times 10^4$ m$^3$/year. The low-flow availability level scenario for the agricultural sector exhibited a shortfall of [18,600.02, 46,224.63] $\times 10^4$ m$^3$/year for the first five-year planning period, that of [0.00, 44,485.34] $\times 10^4$ m$^3$/year for the second five-year planning period, and that of [0.00, 44,929.65] $\times 10^4$ m$^3$/year for the third five-year planning period. Under the medium-flow availability scenario, the shortfall was [0.00, 32,798.50] $\times 10^4$ m$^3$/year for the first five-year planning period, [0.00, 32,172.31] $\times 10^4$ m$^3$/year for the second five-year planning period, and [0.00, 44,929.64] $\times 10^4$ m$^3$/year for the third five-year planning period. Meanwhile, under the high-flow availability scenario, the first five-year planning period deficit was [0.00, 7600.14], whereas those of the second and third five-year planning periods were [0.00, 6748.68] and [0.00, 44,929.64] $\times 10^4$ m$^3$/year, respectively.

During the three planning periods, the industrial, municipal and agricultural sectors in Dehui, Changchun, and Nong'an all exhibited large water deficits compared with those of the other planning areas, with agriculture being the most dominant of the three sectors. Furthermore, the agricultural sector in Nong'an exhibited the largest water deficit. Although in the interval fuzzy two-stage-robust optimization-based water scarcity model, the water allocation scheme was reset to reasonably reduce the quantity of water allocated to sectors with high water yield and the reduction was appropriately employed in sectors with low water yield and high emissions, the resulting upper and lower bounds were slightly more variable, thereby providing a refined solution. The overall water deficit in the Yinma River Basin decreased, which indicated an increase in the extent to which the

water supply of various water-using sectors in the Yinma River Basin can meet the water demand. Moreover, the water allocation intervals for each sector have improved and the water allocation scheme has been optimized.

### 4.3.3. Reused Water Allocation of Different Water Sectors in Each Planning Area Based on the ITSFR Optimization Method

After further fuzzy optimization allocation, the amount of water reused in Yinma River Basin under different available water resources levels is shown in Table 3.

**Table 3.** The amount of reclaimed water under different levels of available water resources in the Yinma River Basin ($10^4$ m$^3$/year).

| Areas | Sectors | h = 1 | h = 2 | h = 3 |
|---|---|---|---|---|
| Panshi | Industrial | [184.40, 486.19] | [298.04, 320.20] | [481.64, 525.54] |
| | Municipal | [0.00, 0.00] | [0.00, 0.00] | [0.00, 0.00] |
| | Ecological | [244.41, 407.58] | [474.42, 526.57] | [384.66, 527.05] |
| | Agricultural | [0.00, 0.00] | [0.00, 0.00] | [0.00, 0.00] |
| Yongji | Industrial | [0.00, 278.03] | [0.00, 306.58] | [0.00, 339.29] |
| | Municipal | [0.00, 0.00] | [0.00, 0.00] | [0.00, 0.00] |
| | Ecological | [0.00, 335.60] | [0.00, 393.04] | [0.00, 423.98] |
| | Agricultural | [0.00, 0.00] | [0.00, 0.00] | [0.00, 0.00] |
| Shuangyang | Industrial | [0.00, 163.88] | [0.00, 189.07] | [0.00, 210.69] |
| | Municipal | [0.00, 0.00] | [0.00, 0.00] | [0.00, 0.00] |
| | Ecological | [0.00, 208.17] | [0.00, 243.33] | [0.00, 274.24] |
| | Agricultural | [0.00, 0.00] | [0.00, 0.00] | [0.00, 0.00] |
| Jiutai | Industrial | [67.91, 368.23] | [102.18, 448.46] | [148.20, 533.29] |
| | Municipal | [0.00, 0.00] | [0.00, 0.00] | [0.00, 0.00] |
| | Ecological | [0.00, 361.67] | [0.00, 418.25] | [0.00, 466.88] |
| | Agricultural | [0.00, 0.00] | [0.00, 0.00] | [0.00, 0.00] |
| Dehui | Industrial | [0.00, 0.00] | [0.00, 0.00] | [0.00, 0.00] |
| | Municipal | [0.00, 0.00] | [0.00, 0.00] | [0.00, 0.00] |
| | Ecological | [327.60, 177.96] | [206.62, 375.35] | [242.27, 414.59] |
| | Agricultural | [0.00, 0.00] | [0.00, 0.00] | [0.00, 0.00] |
| Yitong | Industrial | [0.00, 0.00] | [0.00, 0.00] | [0.00, 0.00] |
| | Municipal | [0.00, 0.00] | [0.00, 0.00] | [0.00, 0.00] |
| | Ecological | [124.09, 143.94] | [115.25, 146.15] | [130.24, 166.04] |
| | Agricultural | [0.00, 0.00] | [0.00, 0.00] | [0.00, 0.00] |
| Changchun | Industrial | [0.00, 2615.72] | [39.06, 1317.84] | [0.00, 2243.61] |
| | Municipal | [0.00, 0.00] | [0.00, 0.00] | [0.00, 0.00] |
| | Ecological | [1548.07, 4733.79] | [4972.06, 5672.73] | [5877.43, 6330.24] |
| | Agricultural | [0.00, 0.00] | [0.00, 0.00] | [0.00, 0.00] |
| Nongan | Industrial | [0.00, 0.00] | [0.00, 46.63] | [0.00, 68.07] |
| | Municipal | [0.00, 0.00] | [0.00, 0.00] | [0.00, 0.00] |
| | Ecological | [208.40, 354.16] | [253.08, 426.59] | [305.35, 478.40] |
| | Agricultural | [0.00, 0.00] | [0.00, 0.00] | [0.00, 0.00] |

As shown in Table 3, after optimized allocation, the water reuse of the Yinma River Basin is mainly allocated to the industrial and ecological environment sectors. In Panshi City, the water reuse of the industrial sector was [184.40, 486.19] $\times$ $10^4$ m$^3$/year, whereas that of the ecological environment sector was [244.41, 407.58] $\times$ $10^4$ m$^3$/year. In the second five-year planning period, the water reuse of the industrial sector was [298.03, 320.20] $\times$ $10^4$ m$^3$/year, whereas that of the ecological environment sector was [474.42, 526.57] $\times$ $10^4$ m$^3$/year. In the third five-year planning period, the water reuse of the industrial sector was [481.64, 525.54] $\times$ $10^4$ m$^3$/year, whereas that of the ecological environment sector was [384.66, 527.05] $\times$ $10^4$ m$^3$/year. In Yongji County, in the first five-year planning

period, the water reuse of the industrial sector was [0.00, 278.03] $\times 10^4$ m$^3$/year, whereas that of the ecological environment sector was [0.00, 335.60] $\times 10^4$ m$^3$/year. In the second five-year planning period, the water reuse of the industrial sector was [0.00, 306.58] $\times$ $10^4$ m$^3$/year, whereas that of the ecological environment sector was [0.00, 393.04] $\times 10^4$ m$^3$/year. In the third five-year planning period, the water reuse of the industrial sector was [0.00, 339.29] $\times 10^4$ m$^3$/year, whereas that of the ecological environment sector was [0.00, 423.98] $\times 10^4$ m$^3$/year. In Shuangyang, in the first five-year planning period, the water reuse of the industrial sector was [0.00, 163.88] $\times 10^4$ m$^3$/year, whereas that of the ecological environment sector was [0.00, 208.17] $\times 10^4$ m$^3$/year. In the second five-year planning period, the water reuse of the industrial sector was [0.00, 189.07] $\times 10^4$ m$^3$/year, whereas that of the ecological environment sector was [0.00, 243.33] $\times 10^4$ m$^3$/year. In the third five-year planning period, the water reuse of the industrial sector was [0.00, 210.69] $\times 104$ m$^3$/year, whereas that of the ecological environment sector was [0.00, 274.24] $\times 10^4$ m$^3$/year. In Jiutai, for the first five-year planning period, the water reuse of the industrial sector was [67.91, 368.23] $\times 10^4$ m$^3$/year, whereas that of the ecological environment sector was [0.00, 361.68] $\times 10^4$ m$^3$/year. In the second five-year planning period, the water reuse of the industrial sector was [102.17, 448.46] $\times 10^4$ m$^3$/year, whereas that of the ecological environment sector was [0.00, 418.25] $\times 10^4$ m$^3$/year. Furthermore, in the third five-year planning period, the water reuse of the industrial sector was [148.20, 533.29] $\times 10^4$ m$^3$/year, whereas that of the ecological environment sector was [0.00, 466.88] $\times$ $10^4$ m$^3$/year. Dehui City only distributes recycled water to the ecological environment sector. The allocation of recycled water in the first, second, and third planning periods was [177.96, 327.60] $\times 10^4$ m$^3$/year, [206.62, 375.35] $\times 10^4$ m$^3$/year, and [242.27, 414.59] $\times 10^4$ m$^3$/year, respectively. Similar to Dehui City, Yitong Manchu Autonomous County only distributes water recycling to the ecological environment sector. The allocation of water recycling in the county for the first, second, and third planning periods was [124.09, 143.94] $\times 10^4$ m$^3$/year, [115.25, 146.15] $\times 10^4$ m$^3$/year, and [130.24, 166.04] $\times 10^4$ m$^3$/year, respectively. In Changchun, for the first five-year planning period, the water reuse of the industrial sector was [0.00, 2615.72] $\times 10^4$ m$^3$/year, and that of the ecological environment sector was [1548.07, 4733.79] $\times 10^4$ m$^3$/year. In the second five-year planning period, the water reuse of the industrial sector was [39.06, 1317.84] $\times 10^4$ m$^3$/year, whereas that of the ecological environment sector was [4972.06, 5672.73] $\times 10^4$ m$^3$/year. In the third five-year planning period, the water reuse of the industrial sector was [0.00, 2243.61] $\times 10^4$ m$^3$/year, whereas that of the ecological environment sector was [5877.42, 6330.24] $\times 10^4$ m$^3$/year. In Nong'an County, only recycling water is allocated to the ecological environment sector, similar to Dehui City and Yitong Manchu Autonomous County. The allocation of recycling water in the first, second, and third planning periods was [208.40, 354.16] $\times 10^4$ m$^3$/year, [253.08, 426.59] $\times 10^4$ m$^3$/year, and [305.35, 478.40] $\times 10^4$ m$^3$/year, respectively.

As shown in Table 3, Changchun, being the capital city, has a relatively prominent issue of water demand. To aid water conservation efforts, Changchun can act to alleviate the deficit of freshwater resources. In view of the conspicuous problem of water deficits in large- and medium-sized cities, measures such as water resource reuse should be taken in the industrial and ecological water consumption sectors. Water deficit situations of different water sectors in various planning periods and areas can be significantly alleviated by employing these measures to adapt to the specific planning period and to urban development. In Changchun area, after optimized allocation of industrial sectors and ecological environment sectors, the water reuse increased yearly from [4163.79, 4733.79] $\times$ $10^4$ m$^3$/year in the first five-year planning period to [5011.12, 6990.57] $\times 10^4$ m$^3$/year in the second five-year planning period. Finally, the third five-year planning period reached [5877.48, 6330.24] $\times 10^4$ m$^3$/year, far exceeding the total water consumption of ecological environment sectors in other planning areas in the same planning period ([2071.04, 2371.18] $\times 10^4$ m$^3$/year, [2343.67, 2946.15] $\times 10^4$ m$^3$/year, and [2671.33, 3449.09] $\times 10^4$ m$^3$/year). In recent years, China has focused on solving its ecological and environmental problems, with a simultaneous increase in industrial water consumption. In view of the particularity

of ecological and environmental water consumption, the reuse of water resource is a good response measure to solve water resource deficit issues. This is especially relevant to areas with relative water shortages and an excessive demand for water resources.

## 5. Conclusions

In this study, an ITSFR optimization model was established to optimize the allocation of water resources and sewage absorption upgrading project in the Yinma River Basin. The model fully considers the fuzzy uncertainty of sectoral water quotas. Additionally, the model can shorten the decision-making interval and ease the decision-making process for the relevant authorities. The economic benefit interval of the Yinma River Basin during the planning period was shortened following optimization, and the economic benefit interval of the full, flat, and dry water periods was shortened by more than 45%, which effectively improved the decision-making efficiency. Upon implementing the project to improve sewage absorption capacity, the amount of sewage discharged into the river by various water-use sectors in the Yinma River Basin decreased. In terms of water resource allocation, the problem of water resource waste caused by high water consumption quota was effectively resolved, whereas the missing water quantity of each water consumption sector was significantly reduced, along with an improvement in the reuse water utilization rate. In summary, the optimization of the ITSFR model can not only improve economic benefits but also shorten the decision-making space. The model also facilitates implementation and other aspects, rendering the overall process more efficient for decision makers.

**Author Contributions:** Conceptualization, W.H. and Z.R.; methodology, W.H. and H.Y.; software, H.Y.; validation, L.Y. and H.Z.; formal analysis, H.Y.; investigation, Z.D.; resources, P.S.; data curation, H.Y.; writing—original draft preparation, H.Z. and M.L.; writing—review and editing, H.Z. and W.H.; visualization, H.Z. and H.Y.; supervision, Z.R.; project administration, S.W. and Y.L.; funding acquisition, H.X., S.W. and Y.L. All authors have read and agreed to the published version of the manuscript.

**Funding:** This research received no external funding.

**Institutional Review Board Statement:** Not applicable.

**Informed Consent Statement:** Not applicable.

**Data Availability Statement:** The data presented in this study are available contained within the article.

**Conflicts of Interest:** The authors declare no conflict of interest.

## Abbreviations

| | |
|---|---|
| $\lambda^{\pm}$ | The membership degree interval of the model. |
| $f_z^{\pm}$ | The economic benefit of ITSR model |
| $f_1^{\pm}$ | Income from water use |
| $f_2^{\pm}$ | Cost of water use |
| $f_3^{\pm}$ | Sewage treatment cost |
| $f_4^{\pm}$ | Pollutant-carrying capacity increases project cost |
| $f_5^{\pm}, f_6^{\pm}, f_7^{\pm}$ | Control the cost of punishment |
| $j$ | Planning area |
| $k$ | Departments that use water |
| $t$ | Planning periods |
| $h$ | The available discharge level of the Yinma River Basin |
| $L_t$ | Length of a period, with each period being five-year long |
| $UNB_{jkt}^{\pm}$ | The unit water resource income of each water-use department |
| $PNB_{jkt}^{\pm}$ | The water shortage loss of each water-use department $k$ in area $j$ in period $t$ ($10^4$ yuan/$10^4$ m$^3$) |
| $p_h$ | The occurrence probability of scenario $h$ |

| $IAW_{jkt}^{\pm}$ | The amount of water resources supplied in advance by the Yinma River Basin to department $k$ in area $j$ in period $t$ ($10^4$ m$^3$/year) |
|---|---|
| $RW_{jkt}^{\pm}$ | The amount of reused water used by department $k$ in area $j$ in period $t$ |
| $DW_{jkth}^{\pm}$ | The lack of water in the Yinma River Basin at level $h$ in period $t$ because it does not meet the water resource distribution plan of department $k$ in area $j$ ($10^4$ m$^3$/year) |
| $CW_{jkt}^{\pm}$ | The water resources use cost of each water-use department $k$ in area $j$ in period $t$ ($10^4$ yuan/$10^4$ m$^3$) |
| $CRW_{jkt}^{\pm}$ | The cost of water reusing for all water-use departments $k$ in area $j$ in period $t$ ($10^4$ yuan/$10^4$ m$^3$) |
| $CWW_{jkt}^{\pm}$ | The cost of sewage treatment for all water-use departments $k$ in area $j$ in period $t$ ($10^4$ yuan/$10^4$ m$^3$) |
| $CRWT_{jkt}^{\pm}$ | The cost of water reuse for all water-use departments $k$ in area $j$ in period $t$ ($10^4$ yuan/$10^4$ m$^3$) |
| $i$ | The 11 water environment control units divided for the Yinma River Basin |
| $r$ | The controlled water pollutant |
| $l$ | Pollutant-carrying capacity improvement project |
| $ER_{ilt}^{\pm}$ | The maximum quantity restriction for project $l$ in units $i$ in period $t$ |
| $Y_{ilt}^{\pm}$ | 0–1 planning parameters, with 0 signifying that project $l$ is not implemented and 1 signifying that project $l$ is implemented |
| $CER_{ilt}^{\pm}$ | The engineering cost of pollution capacity improvement project of each control unit in period $t$ |
| $\alpha_{jkt}^{\pm}$ | The wastewater emission coefficient for department $k$ in period $t$ in area $j$ |
| $\rho$ | The robust coefficient, the values are 0, 0.8, and 1 |
| $\theta_{\mathrm{h}}$ | Relaxation variable |
| $AWQ_{th}^{\pm}$ | The amount of available water resources under level $h$ of period $t$ ($10^4$ m$^3$/year) |
| $WD_{\min jkt}^{\pm}$ | The lowest water consumption quota of various water-use departments $k$ in area $j$ in period $t$ ($10^4$ yuan/$10^4$ m$^3$) |
| $WD_{\max jkt}^{\pm}$ | The highest water consumption quota of various water-use departments $k$ in area $j$ in period $t$ ($10^4$ yuan/$10^4$ m$^3$) |
| $ATW_{jkt}^{\pm}$ | The capacity of sewage treatment of department $k$ in period $t$ in area $j$ ($10^4$ tonnes/year) |
| $\xi_{jkt}$ | The reuse rate of a water department |
| $\beta_{jkt}^{\pm}$ | The sewage discharge coefficient of water-use department $k$ in area $j$ in period $t$ |
| $EC_{krt}^{\pm}$ | The pollutant discharge concentration of the wastewater produced by each water-use department $k$ after centralized treatment in period $t$ (tonnes/$10^4$ m$^3$) |
| $TED_{jrt}^{\pm}$ | The maximum total amount of pollutants in area $j$ in period $t$ (tonnes/year) |
| $IDR_{krt}$ | The inflow coefficient of pollutants discharged by each water-use department $k$ in period $t$ |
| $X_{ij}$ | The discharge coefficient of area $j$ to water environment control unit $i$ |
| $ALD_{irth}^{\pm}$ | The pollutant-carrying capacity of at level $h$ in period $t$ (tonnes/year) |
| $EER_{ilrt}^{\pm}$ | The capacity to enhance the pollutant discharge of the control unit $i$ through the pollutant-carrying capacity improvement project in period $t$ (tonnes/year) |

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
