# Peer review of "Investigating a Water Resource Allocation Model by Using Interval Fuzzy Two-Stage Robust Planning for the Yinma River Basin, Jilin Province, China"

_water, doi:10.3390/w13212974_

Round 1
Reviewer 1 Report
In this paper, a model based on the interval fuzzy two-stage robust method is developed for the optimal allocation of water resources in the Yinma River basin. The results of the model simulation enable the Yinma River basin to be optimized in terms of economic efficiency, pollution absorption capacity and water resources allocation. The article is innovative and has some practical implications, and is recommended for publication after revision.
1. The parameters in section 3.1 are suggested to be organized in a table for reader to read
2. Figure 1 please adjust to a uniform font and convert Chinese characters to English
3. Please harmonize the writing of Figures 2 and 3 with regard to the 5-year planning period/five-year planning period.
4. Could the model constructed in this paper be applied to other watershed plans? If so, what factors should be considered when applying.
5. The form of the brackets in line 906 is inconsistent with the others, please harmonize.
6. A space should be added after the comma for data in interval form in the table.
7. Chapter 4 does not have citations to reflect the reliability of the discussion section.
Reviewer 2 Report
Authors made formidable efforts to investigate a water resource allocation model by using interval fuzzy two-stage robust planning for the Yinma River Basin, Jilin Province, China. The paper is practically sound and then authors require to enhance quality of literature review before any further:
1-There are intelligent computing models to evaluate situation of surface water quality parameters (DO, BOD, and COD). Paying attention to this issue can improve the literature review:
-A Novel Multiple-Kernel Support Vector Regression Algorithm for Estimation of Water Quality Parameters
-Reliability assessment of water quality index based on guidelines of national sanitation foundation in natural streams: integration of remote sensing and data-driven models
-Prediction of water quality parameters using evolutionary computing-based formulations
-Prediction of the five-day biochemical oxygen demand and chemical oxygen demand in natural streams using machine learning methods
3-What is novelty/innovation of this study?
4-Authors need to investigate possibility of uncertainty and reliability analysis in the present study.
Round 2
Reviewer 2 Report
Accept as is